# An Image-Aided Sparse Point Cloud Registration Strategy for Managing Stockpiles in Dome Storage Facilities

Jidong Liu, Seyyed Meghdad Hasheminasab, Tian Zhou, Raja Manish and Ayman Habib *

Lyles School of Civil Engineering, Purdue University, West Lafayette, IN 47907, USA
* Correspondence: ahabib@purdue.edu

**Abstract:** Stockpile volume estimation plays a critical role in several industrial/commercial bulk material management applications. LiDAR systems are commonly used for this task. Thanks to Global Navigation Satellite System (GNSS) signal availability in outdoor environments, Uncrewed Aerial Vehicles (UAV) equipped with LiDAR are frequently adopted for the derivation of dense point clouds, which can be used for stockpile volume estimation. For indoor facilities, static LiDAR scanners are usually used for the acquisition of point clouds from multiple locations. Acquired point clouds are then registered to a common reference frame. Registration of such point clouds can be established through the deployment of registration targets, which is not practical for scalable implementation. For scans in facilities bounded by planar walls/roofs, features can be automatically extracted/matched and used for the registration process. However, monitoring stockpiles stored in dome facilities remains to be a challenging task. This study introduces an image-aided fine registration strategy of acquired sparse point clouds in dome facilities, where roof and roof stringers are extracted, matched, and modeled as quadratic surfaces and curves. These features are then used in a Least Squares Adjustment (LSA) procedure to derive well-aligned LiDAR point clouds. Planar features, if available, can also be used in the registration process. Registered point clouds can then be used for accurate volume estimation of stockpiles. The proposed approach is evaluated using datasets acquired by a recently developed camera-assisted LiDAR mapping platform—Stockpile Monitoring and Reporting Technology (SMART). Experimental results from three datasets indicate the capability of the proposed approach in producing well-aligned point clouds acquired inside dome facilities, with a feature fitting error in the 0.03–0.08 m range.

**Keywords:** stockpile monitoring; LiDAR; feature extraction; feature matching; sparse point cloud; registration; quadratic surfaces/curves





## 1. Introduction

Stockpile volume estimation is quite important for bulk material management. Depending on the type of stockpiles (e.g., aggregate, grain, salt), they could be stored either outdoors or within indoor facilities. Salt, which is quite important for ensuring traffic flow under winter storm conditions, is usually stored within indoor facilities [1]. As shown in Figure 1, barn and dome enclosures are the typical indoor salt stockpile storage facilities. Within the state of Indiana (USA), 60% of the salt storage facilities are domes. In recent years, stockpile volume estimation has been conducted using photogrammetric and LiDAR-based approaches [2–7]. Image-based techniques use automatically identified conjugate points in overlapping images for deriving a 3D model covering the area of interest through *Structure from Motion* (SfM) strategies [8]. These approaches might fail when dealing with datasets in indoor facilities due to unfavorable lighting conditions, as well as the texture-less nature of stockpiles.

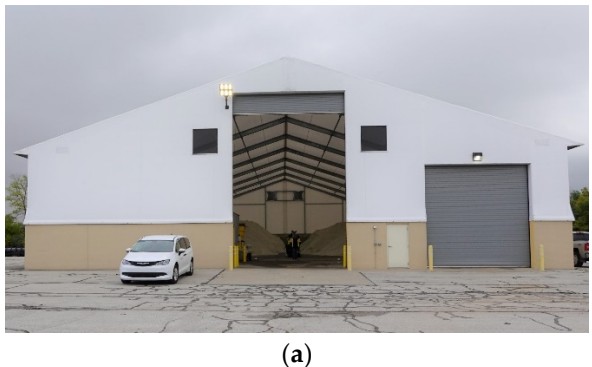 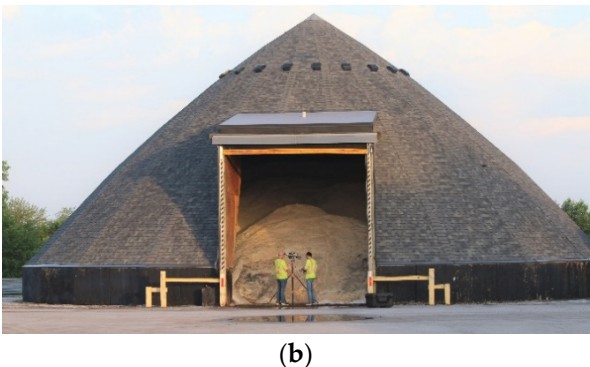

(**a**)                                                                                     (**b**)

**Figure 1.** Samples of salt storage facilities: (**a**) rectangular barn and (**b**) dome.

With recent advancements in LiDAR-based platforms (mobile or static), these systems have been increasingly adopted for direct derivation of 3D point clouds covering stockpiles [9,10]. Mobile LiDAR mapping, e.g., Uncrewed Aerial Vehicles (UAVs), relies on the availability of an integrated Global Navigation Satellite System and Inertial Navigation System (GNSS/INS) onboard the platform to determine the position and orientation (georeferencing parameters) of the ranging sensor. Due to GNSS signal outages, mobile LiDAR cannot be used for mapping indoor stockpiles. Therefore, static LiDAR systems, such as Terrestrial Laser Scanners (TLS), are usually used for monitoring indoor stockpiles [6]. However, due to their high cost and time-consuming data acquisition/processing procedures, using TLS for monitoring indoor stockpiles is not scalable.

As a solution to the abovementioned limitations of imaging and LiDAR mapping technologies, a camera-assisted LiDAR mapping platform, denoted as *Stockpile Monitoring and Reporting Technology* (SMART), has been developed [11,12]. The cost-effective design of the SMART system allows for fast and simple data collection of salt stockpiles with varying sizes and shape complexities. Nevertheless, such hardware design leads to sparse LiDAR point clouds with varying point density and a low percentage of overlap between captured scans. These characteristics need to be considered in the data processing steps.

An important data processing step when dealing with static LiDAR data is aligning (registering) different point clouds to a common reference frame. The registration process aims at estimating the 3D similarity transformation parameters—i.e., three rotation angles, three translations, and a scale factor—between two datasets, denoted as *reference* and *source* point clouds. For a well-calibrated LiDAR sensor, the laser range provides the true scale; thus, the registration parameters reduce to six (scale factor is unity). Registration can be conducted using either point- or feature-based strategies. Point-based approaches assume that the datasets are roughly aligned. The Iterative Closest Point (ICP) is a common approach that uses spatial proximity to establish point pairs between the *reference* and *source* point clouds for the derivation of registration parameters through a Least Squares Adjustment (LSA) procedure [13,14]. Establishing point pairs and estimating the registration transformation parameters are conducted iteratively until the root mean square of distances between conjugate points is smaller than a pre-defined threshold. ICP and its variants have been used in several other studies [15–17]. For example, instead of identifying direct point-to-point correspondences, Al-Durgham et al. [17] assigned each point in the *source* point cloud to a triangular patch (i.e., closest three points) in the *reference* one. Then, the *source* points are projected onto the respective *reference* patches to derive point-to-point correspondences, which are then used for the estimation of registration parameters. ICP and its variants are widely used in several fields dealing with point cloud registration. Cardani et al. [18] utilized ICP to fine-tune the alignment of LiDAR scans collected inside historical masonry vaults for survey and shape computation. Glira et al. [19] performed strip adjustment of *Airborne Laser Scanning* (ALS) data while using several variants of the ICP algorithm. Although point-based techniques can efficiently conduct the registration process, they will fail whenever the initial transformation parameters are not of good

quality and/or there is only a small percentage of overlap between the *reference* and *source* point clouds. Moreover, such methods are sensitive to variations in the point density. For the SMART system, acquired data from different scans have minimal overlap and exhibit significant variation in point density (the nature of acquired point clouds will be shown later in the next section).

Feature-based techniques can reliably estimate the registration parameters while having variation in the point density, without the need for a high percentage of overlap between point clouds, if enough number of conjugate features with a good distribution are identified [20–22]. These algorithms are based on the fact that conjugate features among different LiDAR point clouds would fit a single parametric model after refining the transformation parameters through the LSA strategy. More specifically, the LSA simultaneously solves for the parameters describing the registration transformation function and parametric models of used features (e.g., planar, linear, and/or cylindrical features) through the minimization of the sum of squared normal distances between points belonging to different features and their respective parametric model. Therefore, identifying points that belong to corresponding features is the primary task of these approaches. Some studies conducted feature-based registration by deploying special targets (e.g., highly reflective checkerboards and/or spherical targets) in the study site and then identifying them in different point clouds [23–26]. In order to increase the level of automation, several target-less registration approaches have been proposed that rely on natural geometric features (e.g., planar patches or linear features) in the area of interest [27–31]. For instance, Lin et al. [27] extracted and matched planar, linear, and cylindrical features from point clouds acquired near a bridge. Extracted conjugate features were then used in a LSA engine for the derivation of registration and feature parameters. In the previous work dealing with SMART [11,12], planar features were used for registering point clouds acquired in rectangular salt storage facilities. However, these geometric features do not exist when dealing with stockpiles within dome storage facilities.

This study introduces a registration technique that can deal with data collected by the SMART system inside dome storage facilities. For these facilities, the roof and exposed stringers are extracted and used as quadratic surface and curves, respectively. Moreover, a segmentation strategy that uses normalized polar coordinates rather than Cartesian coordinates of LiDAR points is proposed to handle variations in the point density for extracting planar features, if available.

The remainder of this paper is organized as follows: the SMART system, data collection procedure, and study sites are introduced in Section 2; Section 3 illustrates the proposed data processing framework; experimental results are presented in Section 4; and Section 5 provides the research conclusions and recommendations for future work.

## 2. SMART System and Datasets Description

This section provides a brief introduction to the SMART system and its data collection procedure. In addition, datasets used in this study are introduced.

### 2.1. SMART System Description

The SMART system, as shown in Figure 2, comprises one RGB camera, two LiDAR units, a computer module, and an optional GNSS unit. The RGB camera is a GoPro Hero 9 that has a 5184 × 3888 CMOS array with a 1.4 μm pixel size and a lens with a nominal focal length of 3 mm. Two Velodyne VLP-16 Puck LiDAR sensors with orthogonal orientation are installed to increase the LiDAR units' field of view. Such design allows for a broader coverage of the area, which facilitates the derivation of required features for point cloud registration. Each LiDAR unit consists of 16 radially oriented laser beams that are vertically aligned from −15° to +15° and capable of rotating 360° internally in the horizontal direction. The VLP-16 Puck has a maximum measurement range of 100 m and a range accuracy of ±3 cm. The computer module is a Raspberry Pi 4b that initiates the LiDAR data acquisition and stores collected point clouds. The system is equipped with a GNSS antenna/receiver

for the georeferencing of acquired point clouds when operating in outdoor environments. As shown in Figure 2, in order to reduce occlusions in the captured data, the SMART system is deployed in the storage site using either an extendable tripod with a maximum height of 6 m (Figure 2a) or through permanent mounting on the roof (Figure 2b).

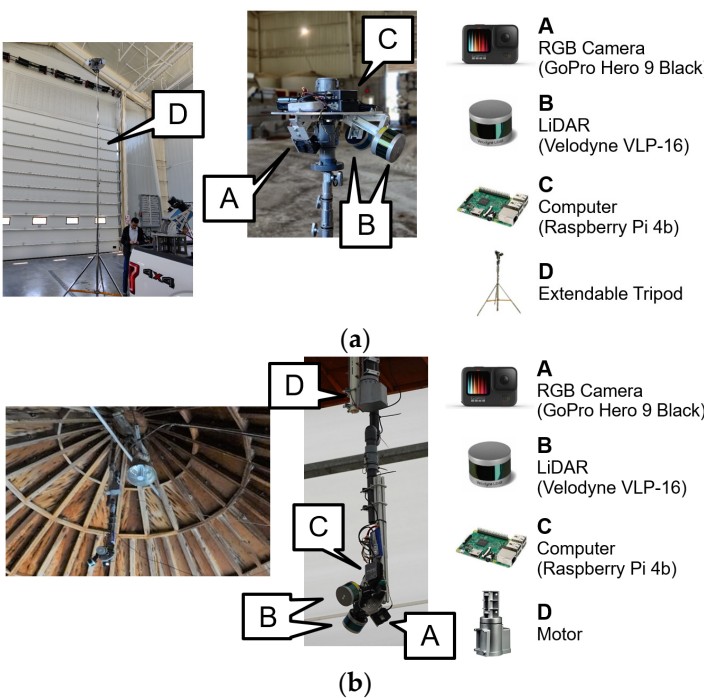

**Figure 2.** The integrated SMART system used in this study with two configurations: (**a**) tripod-mounted and (**b**) roof-mounted.

A system calibration procedure was conducted as described in [11] to estimate the individual sensor parameters, as well as the mounting parameters relating different sensors. Camera Interior Orientation Parameters (IOP) including principal distance ($c$), principal point coordinates ($x_p$, $y_p$), and radial/de-centering lens distortions ($K_1$, $K_2$, $P_1$, $P_2$) were estimated using a test field comprised of several checkerboard targets with known distances between some of the targets. For the mounting parameters, a pole coordinate system was first defined to establish the relative position/orientation of onboard sensors, as depicted in Figure 3. Then, the mounting parameters between individual sensors and the pole coordinate frame were derived through a LSA process by minimizing discrepancies among conjugate LiDAR and image features.

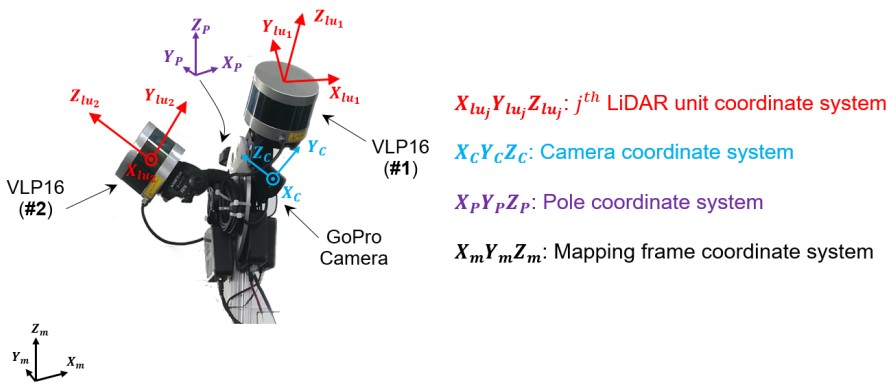

**Figure 3.** Illustration of the involved coordinate systems in the SMART system.

## 2.2. SMART System Operation

At each instance of data collection, hereafter referred to as a scan, the SMART system captures a pair of LiDAR point clouds, along with one RGB image, as shown in Figure 4. As can be seen in this figure, only a portion of the stockpile is covered by the point cloud captured at one scan. To have as much coverage as possible, system motion is needed to acquire multiple scans. The SMART system motion is realized by rotating the pole manually or mechanically around its vertical axis in approximately 30° increments, as illustrated in Figure 5a. Given this rotation angle, a minimum of 7 scans are required to ensure a 360° horizontal field of view by the two LiDAR sensors. In practice, the number of scans can vary between 7 to 13. The SMART system produces sparse LiDAR scans with significant variation in point-density due to this simple and cost-effective design and data acquisition procedure. Additionally, there is insufficient overlap between successive scans, as shown in Figure 5b.

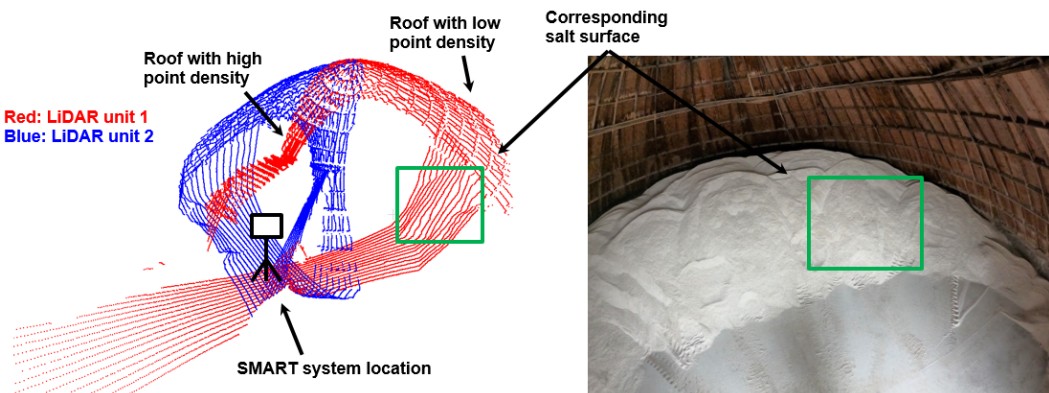

**Figure 4.** A sample point cloud (colored by LiDAR unit ID) and image captured by the SMART system at a given scan.

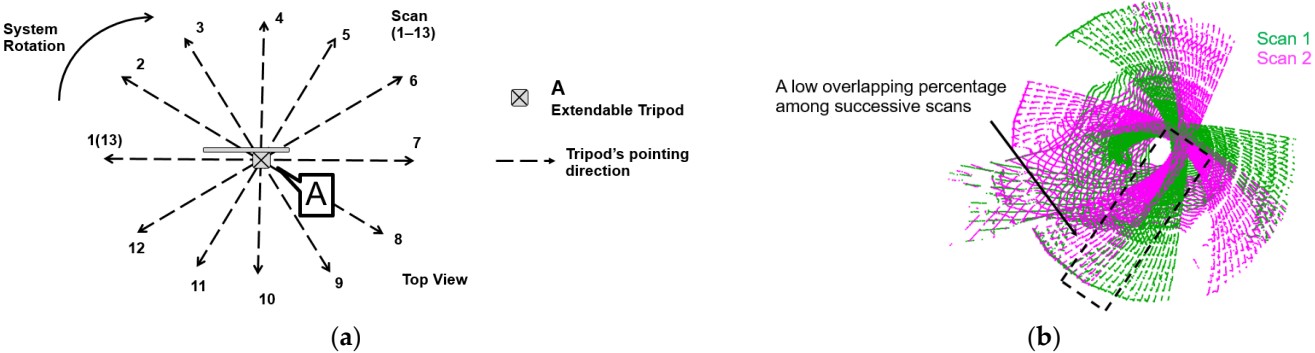

**Figure 5.** Data collection procedure for the SMART system and acquired LiDAR scans: (**a**) seven to thirteen scans are acquired at a given station and (**b**) limited overlap area between two successive scans.

## 2.3. Datasets Description

In this study, three datasets collected within dome salt storage facilities managed by the *Indiana Department of Transportation* (INDOT) are used to evaluate the performance of the proposed registration strategy. These facilities (shown in Figure 6) are located in Lebanon, Frankfort, and West Lafayette, Indiana, USA. One should note that due to the excessive amount of salt stockpile in the *Frankfort* dataset, the SMART system was located at the entrance of the facility, while in the other two datasets, the system was placed close to the dome center (see Figure 6). The impact of the SMART location within the dome will be explained in the methodology and experimental results sections. Table 1 lists the facility size, number of acquired scans, and location of the SMART system within the facility.

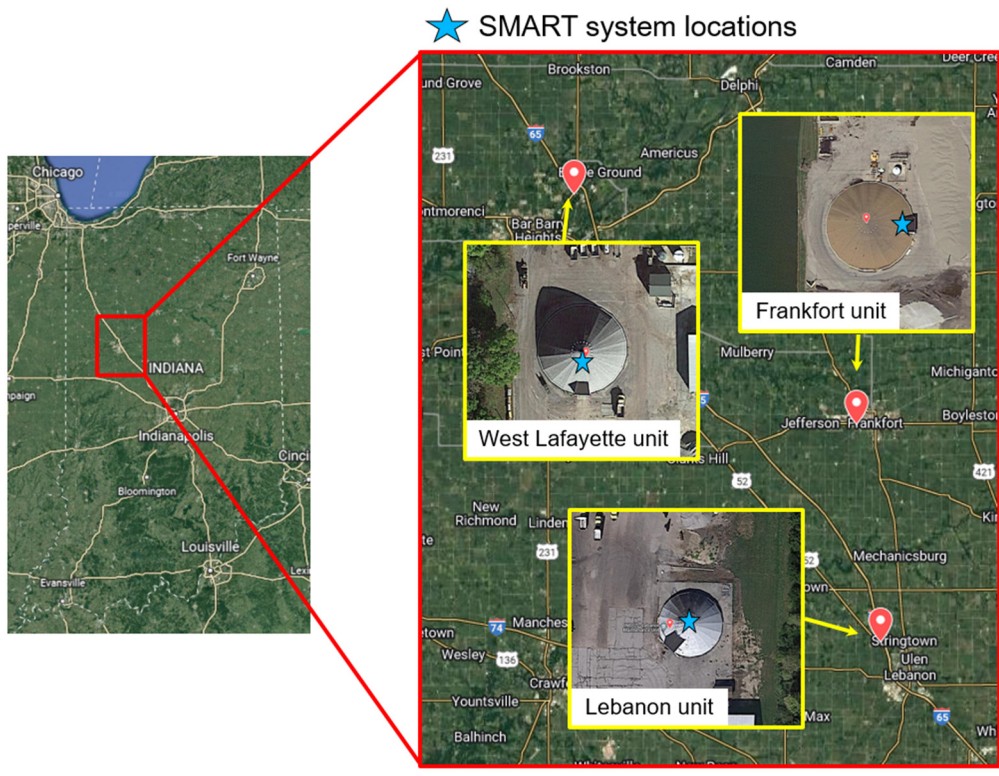

**Figure 6.** Locations of the three salt dome facilities used in this study along with a sample aerial view of each facility (aerial photo adopted from Google Earth Image).

**Table 1.** Data collection sites, facility size, number of scans, and system location.

| Salt Dome Facility | Size (m) | | Number of Scans | System Location |
|---|---|---|---|---|
| | Radius | Height | | |
| Lebanon unit | 10.0 | 11.0 | 9 | Inside the facility |
| Frankfort unit | 14.5 | 17 | 12 | By the entrance |
| West Lafayette unit | 13.5 | 15.5 | 13 | Inside the facility |

## 3. Methodology

The SMART system delivers a set of point clouds, denoted as a scan, captured by the two LiDAR units after applying incremental rotation angles. The configuration/specifications of the LiDAR units and approximate 30° incremental rotation result in sparse scans with minimal overlap and significant variation in point density (as has been shown in Figures 4 and 5). Such characteristics make the registration problem a challenging one. For a target-less registration procedure, we have to rely on existing features in the storage facility. Given the texture-less nature of the salt stockpile surface, physical features constituting the dome storage facility (roof, roof stringers, entrance walls, as can be seen in Figure 7) are the only remaining alternatives. In general, SMART would acquire scans with sufficient points for reasonable representation of the dome roof (i.e., a sparse set of points would be adequate to represent the quadratic surface of the roof). However, depending on the amount of salt and SMART mounting configuration (tripod-mounted or roof-mounted), captured scans might not provide sufficient points to represent the entrance walls and/or roof stringers. For situations where SMART is set-up close to the center of the dome using either the tripod or permanent mounts, as shown in Figure 7a,b, acquired scans would have sufficient points for the extraction and modeling of roof stringers. One can observe by looking at Figure 7b that the stringers converge towards the dome apex. Alternatively,

when using a tripod-mounted SMART in an almost full facility, we only have sufficient points to represent/model the entrance walls, as shown in Figure 7c.

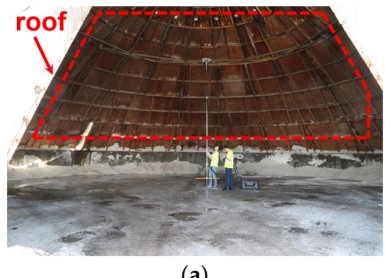 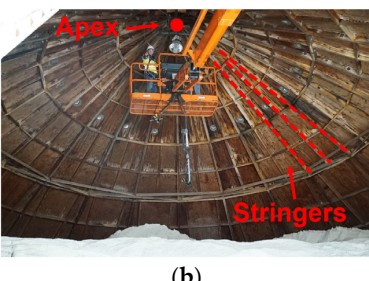 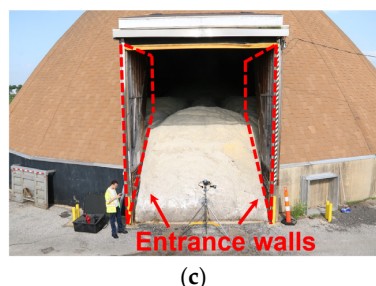

(**a**)　　　　　　　　　　　　　　(**b**)　　　　　　　　　　　　　　(**c**)

**Figure 7.** Demonstration of physical features constituting the dome storage facilities: (**a**) roof, (**b**) roof stringers, and (**c**) entrance walls (the SMART system is tripod-mounted at the center, roof-mounted at the center, and tripod-mounted at the entrance in these three examples, respectively).

At a high level, the proposed registration procedure aims at automated extraction and matching of features (i.e., roof, roof stringers, entrance walls, ground) in the acquired scans. These features are then used to solve for the registration problem by estimating the transformation parameters that minimize the sum of squared normal distances between post-aligned points from different scans and the respective parametric model for the used features. Thus, the unknowns of the LSA optimization include the transformation parameters for the different scans, as well as the parametric model of the used features. The registration framework encompasses three processing blocks, as can be seen in Figure 8. The first block aims at estimating the coarse registration parameters between acquired scans. Specifically, successive images are used to roughly estimate the incremental rotation angle between successive scans. The fine registration is achieved through a two-step procedure, which is represented by the last two blocks in Figure 8. In the first step, as represented by the second block in Figure 8—partial fine registration, extracted and matched roof segments in different scans are used to evaluate a subset of the registration transformation parameters (i.e., shifts in the planimetric and vertical directions and two rotation angles for levelling the point cloud, which are the rotation angles to align the dome axis along the plumb line). Due to the axisymmetric configuration of the roof, the rotation around the dome axis cannot be estimated in this step. This rotation angle is then estimated using extracted/matched features that belong to roof stringers and/or entrance walls (represented by the third block in Figure 8—full fine registration). The remainder of this section provides more details about each of the processing blocks shown in Figure 8.

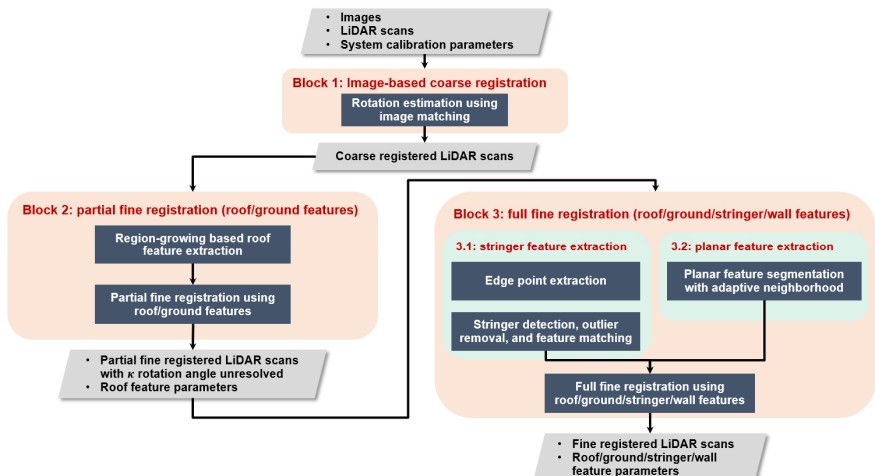

**Figure 8.** Proposed data processing framework for registration of SMART scans acquired in dome facilities.

### 3.1. Coarse Registration of LiDAR Scans

This step aims at coarsely aligning the acquired *S* LiDAR scans with *S* ranging from 7 to 13 (depending on whether 180° or 360° rotation of the SMART mount is adopted). Assuming that the SMART pole coordinate system does not shift when acquiring data, the coarse registration parameters would only involve the estimation of the rotation angle between successive scans. Due to the sparsity of the acquired point clouds, as well as insufficient overlap between successive scans, it is difficult to use LiDAR-based features for coarse registration. To overcome this challenge, an image-aided coarse registration approach was developed, as described in [11]. Assuming that the perspective center of the used camera is spatially close to the SMART rotation axis, automatically established conjugate points between successive images are used to sequentially estimate the incremental rotation angles. To make the process of conjugate point identification more robust, a constrained search space for corresponding points is established using the nominal incremental rotation angle (i.e., 30°). Together with the camera-to-pole and LiDAR-to-pole mounting parameters, derived camera rotations are then used to coarsely align successive LiDAR scans. Figure 9 illustrates the image-based coarse registration process for two scans in the *West Lafayette* dataset. In this figure, the before-coarse registration scans are defined using the nominal incremental rotation angle, while the post-coarse registration scans are defined using the image-based estimate of the rotation. Interested readers can refer to [11] for more details about the mathematical details of the quaternion-based approach for estimating the incremental rotation using identified matches.

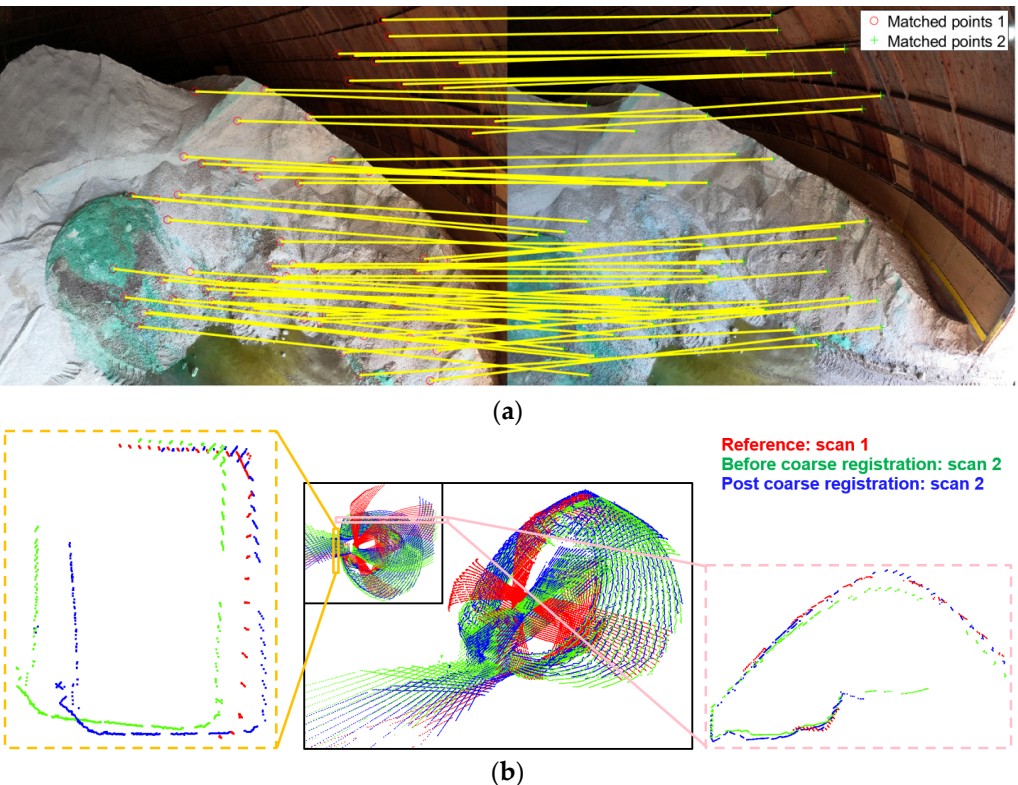

(**a**)

(**b**)

**Figure 9.** Image-aided coarse registration results for two successive scans in the *West Lafayette* dataset: (**a**) identified conjugate points between captured images in these two scans and (**b**) comparison of the before/post coarse registration results through a perspective view (middle) and two vertical profiles of the facility entrance (orange dotted box on the **left**) and the facility roof (pink dotted box on the **right**).

### 3.2. Partial Fine Registration

Given the coarse registration parameters, a feature-based, partial fine registration procedure is conducted using extracted roof surface in the different scans (Figure 10). For roof extraction, this study develops a region-growing strategy starting from interactively selected seed points in the different scans. The roof feature in the different scans are then used for the partial fine registration of acquired point clouds. The roof surface extraction and partial fine registration steps are explained below.

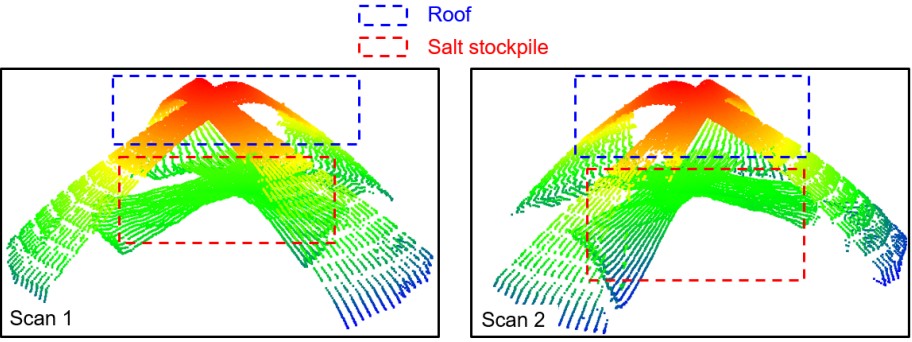

**Figure 10.** Examples of two successive scans colored by height (for easier illustration, the ground has been removed).

### 3.2.1. Roof Extraction

The roof surface is extracted using a region-growing strategy. A traditional region-growing strategy starts with a seed point, which is used to define a seed region (i.e., a local neighborhood of the seed point). The seed region is then used to identify the nature of the parametric model representing that region (e.g., planar or linear local neighborhood). The parameters of that model are derived through a LSA using the constituent points of the seed region (i.e., estimating the parameters that minimize the sum of squared normal distances between the points within the seed region and fitted parametric model). Then, neighboring points to the seed region that belongs to the parametric model are augmented into the region. This process is repeated until no more points can be augmented [32]. Neighborhood definition (either for seed region definition or identification of candidate points to be augmented into the seed/previously segmented region) can be established through a search radius or nearest $K$ points [33].

In this study, a new segmentation procedure is proposed. For the roof, which would be eventually represented by a quadratic surface, we cannot define a reliable initial model of the quadratic function due to the relatively small size of the seed region (i.e., the relatively low curvature of the roof would lead to the definition of planar rather than quadratic surface through the seed region). Other factors that would make the region growing process more challenging include the sparse nature and significant variation in the point density of the SMART scans. To mitigate these challenges, the proposed roof extraction strategy does not use a quadratic surface as the parametric model for the seed region definition and its augmentation. Rather, it starts with identifying a seed region that is relatively planar. Moreover, a hybrid neighborhood definition strategy, using both neighboring points within a search radius and nearest $K$ neighbors, is adopted to deal with the sparsity and density variation within the points constituting the scan.

Figure 11 shows the proposed steps for the roof segmentation strategy. In the first step, given an interactively selected seed point on the roof, a seed region is derived through a user-defined search radius, $r_{seed}$, which is set in a way where the defined seed-region can be assumed to be planar, e.g., $r_{seed} = 1$ m. In Figure 11, the seed region is defined by the green points. Then, a region growing is conducted through a two-stage procedure. First, for each point in the seed region, we identify the $K$ nearest neighbors, while excluding points that belong to the seed region. The accumulated neighbors are considered as potential candidates for the augmentation (blue points in step 2 of Figure 11). Then, for each point in the

potential candidate list, we derive neighboring points, which belong to the seed/previously segmented region within a search radius $r_{seed}$. If the normal distance between the potential candidate point in question and the local plane defined by neighboring points is less than a predefined threshold, the potential candidate point is augmented to the roof surface (steps 3 and 4 in Figure 11). The region growing process (steps 2, 3, and 4) is repeated until no more points can be added to the roof segment. A sample of segmented roof points for a LiDAR scan is shown in Figure 12. Since there is a single roof, we do not need to conduct a matching process for extracted roof surfaces in the different scans.

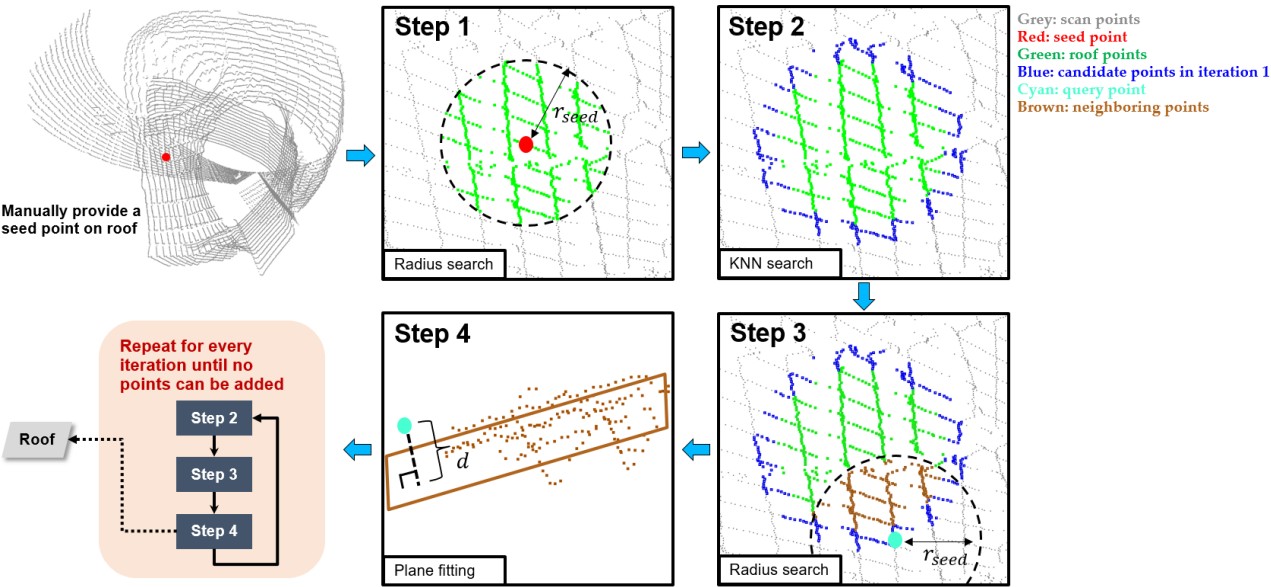

**Figure 11.** Schematic workflow of the proposed roof feature extraction procedure.

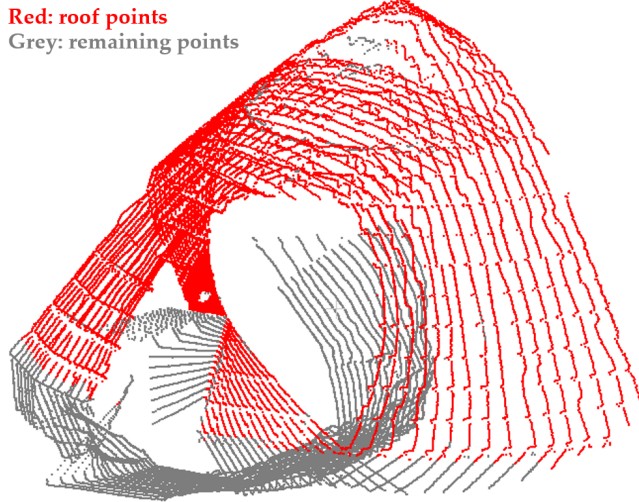

**Figure 12.** A sample roof segmentation result for a LiDAR scan in the *West Lafayette* dataset.

### 3.2.2. LSA Using Roof Feature

The fine registration of the individual scans aims at estimating the position, $r^m_{p(k)}$, and orientation, $R^m_{p(k)}$, of their local/pole coordinate systems relative to a common mapping frame (where $k$ is the scan index). The post-registration coordinates of point $I$ in the $k^{th}$ scan, $r^m_I$, can be derived through Equation (1).

$$r^m_I = r^m_{p(k)} + R^m_{p(k)} r^p_{lu_j} + R^m_{p(k)} R^p_{lu_j} r^{lu_j(k)}_I \tag{1}$$

where:

- $r_I^{lu_j(k)}$ is the coordinates of point $I$ with respect to the $j^{th}$ LiDAR unit coordinate system at the $k^{th}$ scan;
- $r_{lu_j}^p$, $R_{lu_j}^p$ are the mounting parameters (lever arm and boresight rotation matrix) for the $j^{th}$ LiDAR unit relative to the pole coordinate system;
- $r_{p(k)}^m$, $R_{p(k)}^m$ are the translation vector and rotation matrix of the pole coordinate system at the $k^{th}$ scan relative to the mapping frame.

Post registration, the quadratic surface model of the roof is defined by the coordinates of the dome apex $(x_0, y_0, z_0)$ and coefficients $(a, b)$ of the quadratic surface (refer to Figure 13). The $Z$-axis of the mapping frame is defined to be parallel to the dome axis (i.e., registered scans would lead to levelled point clouds). Thus, levelling the scans would help in defining two of the rotation angles relating the coordinate systems for the individual scans and mapping frame. The quadratic surface is defined by the coefficients $(a, b)$ using a dome coordinate system parallel to the mapping frame with its origin at the apex (Figure 13), as shown in Equation (2).

$$F_{roof} = z_d - ar^2 - br = z_d - a\left(x_d^2 + y_d^2\right) - b\sqrt{x_d^2 + y_d^2} = 0 \tag{2}$$

where $(x_d, y_d, z_d)$ represent the transformed scan coordinates of the points along the roof surface after being shifted to the dome coordinate system (i.e., $[x_d, y_d, z_d]^T = [x_m, y_m, z_m]^T - [x_0, y_0, z_0]^T$, with $(x_m, y_m, z_m)$ defined by Equation (1)).

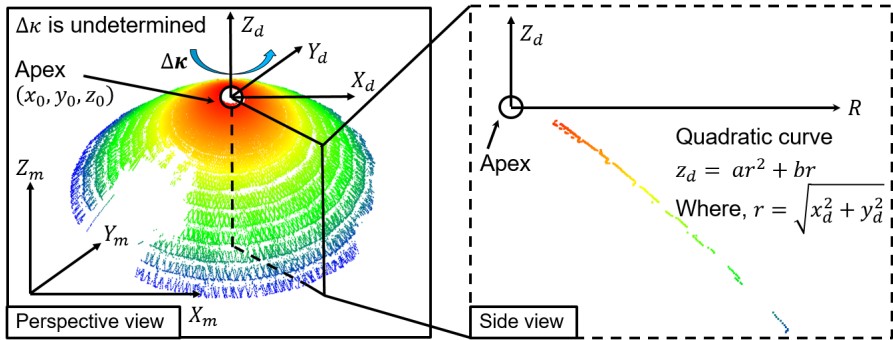

**Figure 13.** Coordinate system definition and parametric model for the roof feature with the axisymmetric characteristic of the roof feature (**left**) together with a vertical profile through the roof (**right**).

Each point on the roof (quadratic surface) provides one constraint of the form in Equation (2). Considering Equations (1) and (2), one can infer that the constraints for the roof feature incorporate the parameters describing dome surface $(x_0, y_0, z_0, a, b)$, as well as registration parameters for the different scans ($R_{p(k)}^m$ and $r_{p(k)}^m$). The LSA uses the constraint equation to solve for the unknown parameters by minimizing the sum of squared deviations between transformed points and the fitted quadratic surface model. As already mentioned, due to the axisymmetric dome surface, the third rotation angle around the $Z$-axis of the mapping frame cannot be defined. Therefore, in the LSA, the initial value for this rotation angle, which is derived from the image-based coarse registration, is not solved for.

### 3.3. Full Fine Registration

So far, all the fine registration parameters are estimated, except for the $\Delta\kappa$ rotation angles for the different scans. To determine these rotation angles, we need to incorporate additional features, such as roof stringers and/or entrance walls. For acquisition scenarios where the SMART is mounted close to the dome center, roof stringers will be more common

in captured scans than entrance walls. The latter will only be common for scenarios where the SMART is mounted close to the dome entrance. The next subsections address the extraction, matching, and incorporation of these features in the LSA to solve for the full fine registration parameters.

3.3.1. Extraction and Matching of Roof Stringers

The stringers will be extracted from the previously segmented roof features from the scans when the SMART system is located close to the center of the facility, e.g., tripod-mounted in Figure 7a or ceiling-mounted in Figure 7b. The stringers follow the shape of the dome surface. Therefore, they can be modeled as quadratic curves. However, their projection onto a horizontal plane defined by the levelled scans, following the partial fine registration, would define straight lines that converge towards the apex. These characteristics of the stringers are used for their extraction and matching. Specifically, a four-step data processing workflow is implemented to derive conjugate stringers among acquired scans, as depicted in Figure 14.

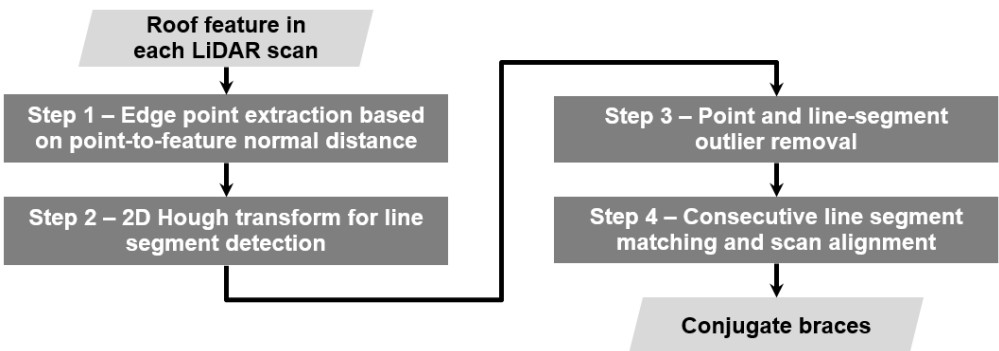

**Figure 14.** Workflow for the extraction and matching of roof stringers in SMART scans.

*Step 1* aims at identifying points within the extracted roof surface that belong to stringers (hereafter, these points will be denoted as edge points). Edge points will be extracted as those that have considerable negative normal distance when compared to their local neighborhood, relative to the defined quadratic surface of the dome (one should note that points below the quadratic surface will have negative normal distance). To initiate the edge point extraction, we start by deriving the signed normal distance, $d_r$, between a query point and the quadratic roof surface (estimated in the previous step). Deriving $d_r$ in a 3D space can be reduced to a 2D point-to-quadratic curve distance estimation by considering a vertical plane that passes through the dome apex and query point, as shown in Figure 15. Then, the normal distance for the query point $(r_q, z_q)$ can be estimated through an optimization procedure that determines its projection $(r_n, z_n)$ onto the quadratic curve in Figure 15 (one should note that the quadratic curve of the vertical slice through the query point and dome quadratic surface will have the same parametric coefficients). The used function for the minimization process is represented by a third order polynomial in $r_n$, as per Equation (3). Following the determination of the normal projection of the query point, the normal distance $d_r$ can be derived as $\sqrt{(r_q - r_n)^2 + (z_q - z_n)^2}$. The signs of the coordinate differences, $(r_q - r_n)$ and $(z_q - z_n)$, are used to determine whether the query point is above or below the quadratic surface. Edge points are established as those with significant negative normal distance when compared to their local neighborhoods. Specifically, given a query point, we define a local neighborhood within a search radius. The signed normal distances in this local neighborhood are sorted in an ascending order. A query point that has a normal distance smaller than the 85th percentile of those in the

local neighborhood will be considered an edge point (i.e., it belongs to a roof stringer). An example of extracted edge points from the roof feature for one scan is shown in Figure 16.

$$
\begin{aligned}
d_r^2 &= (r_q - r_n)^2 + (z_q - z_n)^2 \\
d_r^2 &= (r_q - r_n)^2 + (z_q - ar_n^2 - br_n)^2 \\
\phi(r_n) = \partial d_r^2 / \partial r_n &= (r_q - r_n) + (z_q - ar_n^2 - br_n)(2ar_n + b) = 0
\end{aligned}
\tag{3}
$$

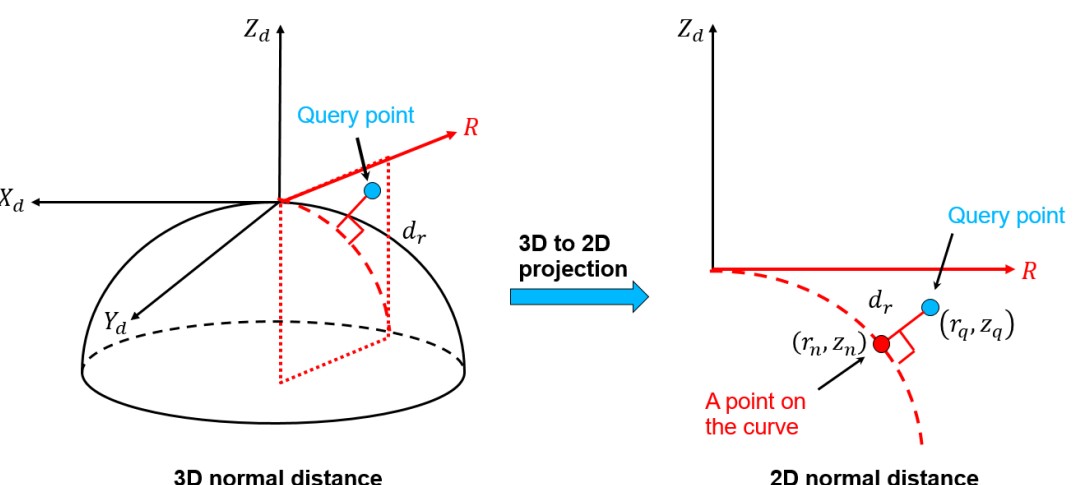

**Figure 15.** Schematic diagram of point-to-quadratic surface distance (**left**) and its 2D projection (**right**).

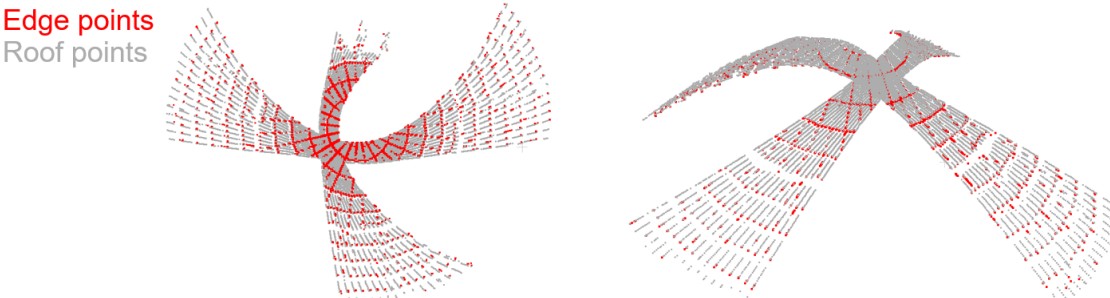

**Figure 16.** A sample edge point detection result for a roof feature in one scan (**left**: top view, **right**: perspective view).

In *Step 2*, we aim at clustering edge points that belong to a distinct roof stringer. First, edge points are projected to a 2D space orthogonal to the defined dome axis by the partial fine registration procedure, as shown in Figure 17a. A 2D Hough transform is then used to detect 2D lines [34]. In order to detect roof stringers on both sides of the apex, which might be collinear, as different 2D lines, the range of the angular line parameter in Hough space is defined as [0°, 360°). Moreover, since we are only looking for linear features that pass through the apex, we only consider detected lines with zero normal distance from the origin of the dome coordinate system. A sample line detection result is shown in Figure 17b. As can be seen in this figure, detected lines include correctly extracted stringers, partially correct stringers due to some outlier points along the stringer, and fully erroneous stringers that do not pass through the roof apex (those could be erroneously detected owing to the cell size of the Hough space accumulator array). Sample stringer detection outliers are shown in Figure 17c.

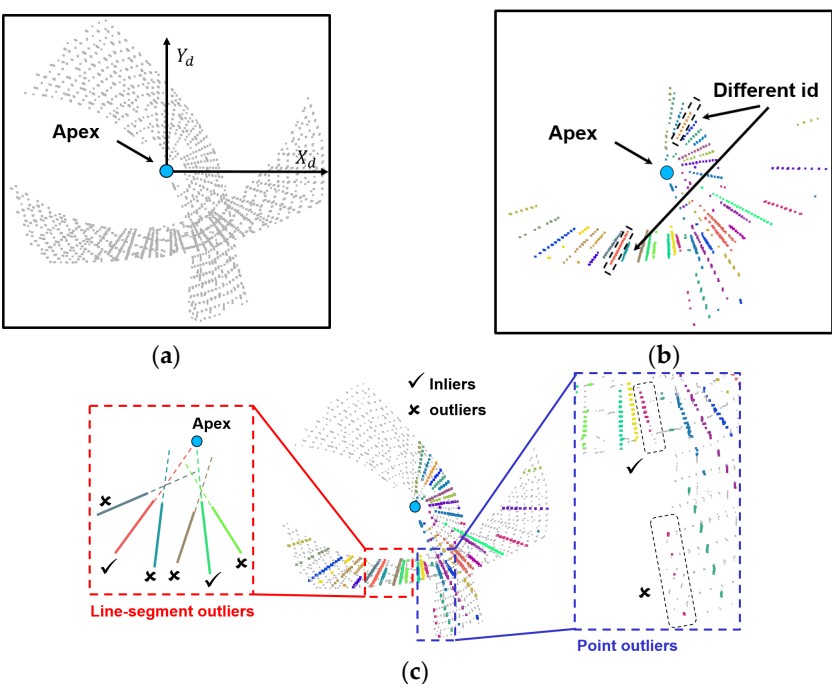

**Figure 17.** An example of 2D line segment detection results: (**a**) projected edge points, (**b**) detected 2D line segments (colored by id), and (**c**) point and line-segment outliers.

In *Step 3*, existing point and line-segment outliers are removed sequentially. First, assuming that for a partially-correct stringer, the majority of the identified edge points are inliers, those minor outlier points are detected and removed. To do so, a distance-based region growing is conducted for grouping the points using spatial proximity as the grouping criterion for each stringer. The group with the maximum number of points is considered as the main cluster. Remaining clusters that exhibit a large angular deviation from the main cluster, e.g., $\geq 1°$, are removed. A sample of identified outlier points is shown in Figure 18a. For fully erroneous stringers, detected lines that do not pass through the roof apex are identified. In the implemented approach, an exhaustive sample consensus search is conducted to derive the best-fitted apex point. Specifically, in each trial, two stringers are selected, and an apex point is derived through their intersection. Consensus among the remaining stringers is evaluated by identifying lines with small normal distance, e.g., $\leq 0.05$ m, from the derived apex point. Once the exhaustive search is completed, the sample with the largest consensus set is used to derive the best-fitted apex. Detected segments with a large apex-to-line distance are removed as outliers. Figure 18b illustrates sample results from the fully-erroneous stringer removal strategy.

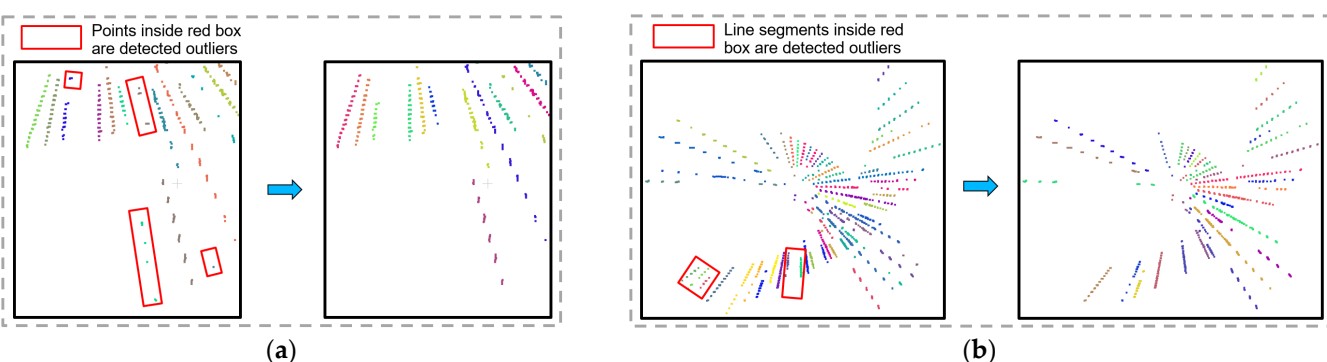

**Figure 18.** Examples of removed outliers: (**a**) point outliers (**left**: before, **right**: after) and (**b**) line-segment outliers. Line segments are colored by id (**left**: before, **right**: after).

Finally, in *Step 4*, a feature matching procedure is conducted to identify conjugate stringers among different scans. Based on the proposed extraction strategy, derived stringers and their 2D projection intersect at a single point (i.e., roof apex). Thus, the azimuth of detected stringers relative to a reference direction following the coarse and partial fine registration of the individual scans can be used as the matching criterion (as shown in Figure 19a). Due to the sequential nature of the image-based coarse registration (which is the only available information for establishing the rotation around the dome axis), the azimuth evaluation for the stringers in the different scans will exhibit an error propagation that will increase as we move away from the first, *reference* scan. In other words, the azimuth-based stringer matching will only be reliable when considering two successive scans. To avoid the impact of error propagation when matching stringers in non-successive scans, we use matched stringers in successive scans to redefine the stringers' azimuth relative to a new reference direction for the last considered scan (this concept is illustrated in Figure 19b). One should note that the stringer extraction and matching is conducted in 2D, which can be traced backward to the respective ones in 3D. Figure 20 shows established conjugate stringers for the *West Lafayette* dataset. A summary of the stringer extraction steps is provided in Figure 21 (steps 1–4 in Figure 14).

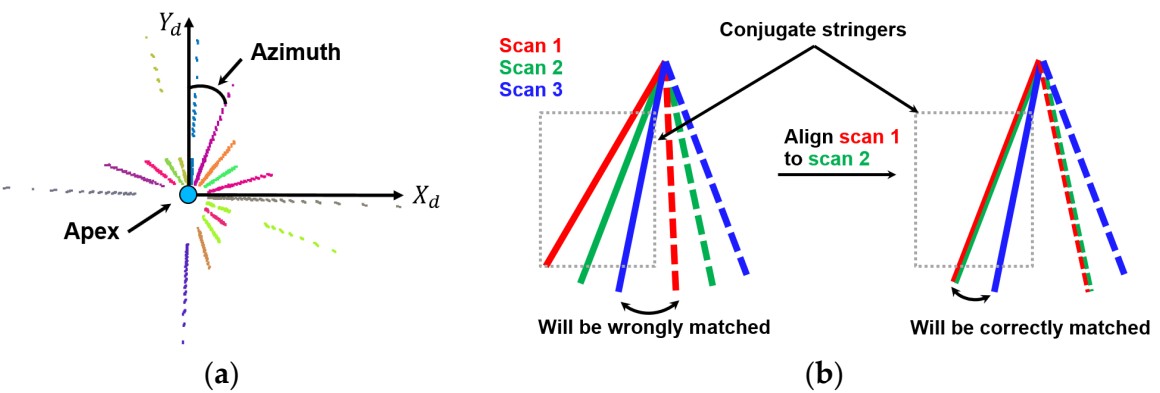

(a)  (b)

**Figure 19.** Illustration for the stringer matching: (**a**) definition of the stringer azimuth and (**b**) the successive matching procedure for mitigating the error propagation.

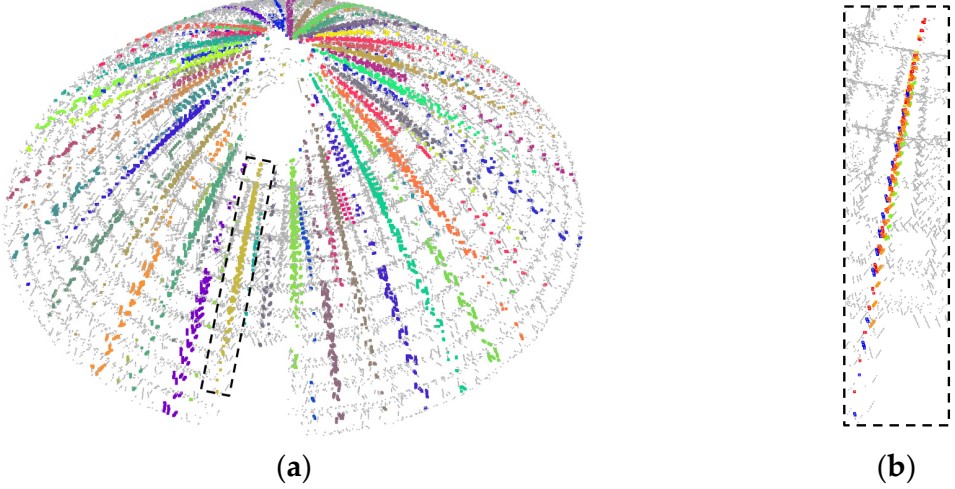

(a)  (b)

**Figure 20.** Established conjugate stringers for the *West Lafayette* dataset: (**a**) conjugate stringers from 13 scans (colored by stringer ID) and (**b**) a zoomed-in region for one stringer (colored by scan ID) (edge points are colored with grey).

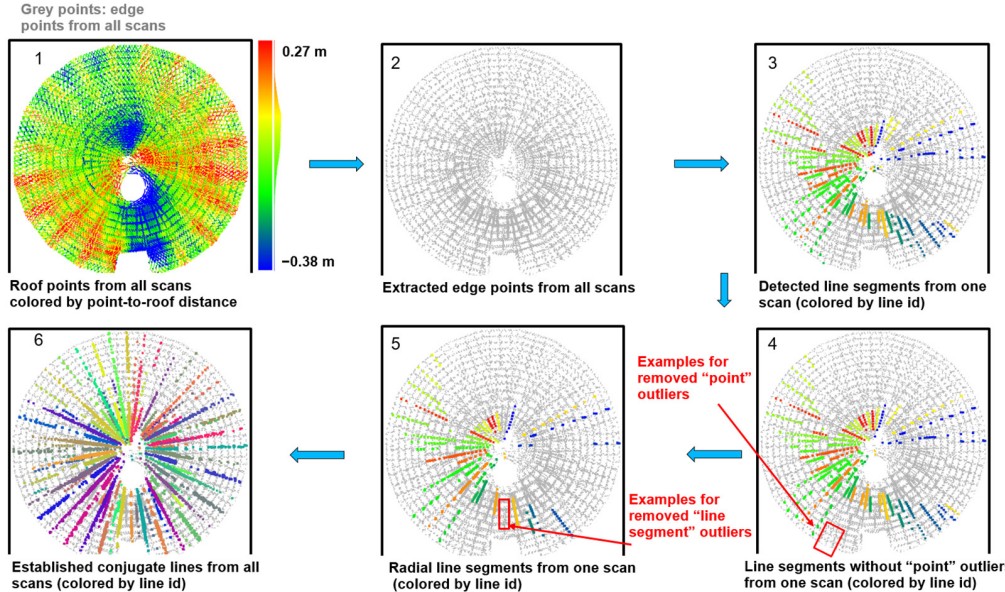

**Figure 21.** A review of the procedures involved in generating conjugate stringers for the *West Lafayette* dataset (all sub-images are shown in top view).

### 3.3.2. Extraction and Matching of Planar Features

The SMART system might sometimes be placed at the entrance of the dome facility (or even outside) due to the high volume of the stored stockpile (as can be seen in Figure 22a). In this case, we might not have enough points to identify stringers (edge points) along the dome surface. For those situations, planar features, i.e., vertical walls and ground, located by the facility's entrance can be used for refining the registration parameters. An example of these features is shown in Figure 22b. It should be noted that although ground features do no provide control for refining the rotation angle around the dome axis, they can be used to ensure accurate parameter estimates for levelling the scans, as well estimating the shift among the scans in the vertical direction.

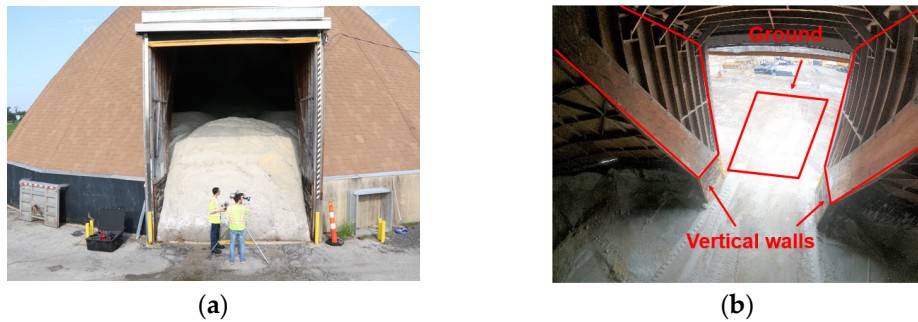

**Figure 22.** Demonstration of planar features near the entrance of a salt dome facility: (**a**) tripod-mounted SMART system stationed near the entrance and (**b**) wall/ground features.

For planar feature extraction from the individual scans, a region growing-based segmentation strategy, which can handle the sparsity and point density variation, is developed. The developed strategy is a modified version of the multi-class simultaneous segmentation (MCSS) approach developed by Habib and Lin [32]. In the MCSS, seed points are first randomly generated in a scan. Then, seed regions are established using neighbors of the seed points. Principle component analysis (PCA) [35] is then carried out to identify seed regions that are planar. Next, a distance-based region growing is conducted to augment coplanar neighboring points to the feature in question. The MCSS modification starts by representing the LiDAR points' neighbors using their normalized polar coordinates rather

than Cartesian ones. Normalized polar coordinates will help in defining more meaningful neighborhoods that are robust to variations in point density. Figure 23 depicts a scan using the Cartesian and normalized polar coordinates of its constituent points. The segmentation results for this scan using these two coordinates are shown in Figure 24. In this figure, one can see that the segmentation result using the normalized polar coordinates has better quality extracted features, especially when the feature has large point density variation (which is the case for the ground feature—refer to Figure 24b). For features with almost uniform point density (e.g., entrance walls), both approaches yield similar results—refer to Figure 24c. Given the available partial fine registration parameters, a feature matching using surface normal orientation and spatial proximity criteria is conducted to derive conjugate planes among scans.

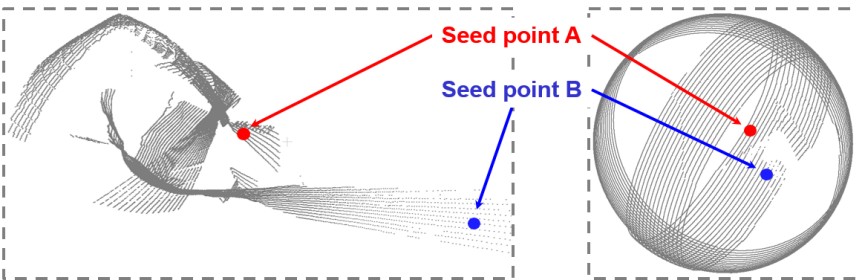

**Figure 23.** A LiDAR scan and two sample seed points; points are represented by Cartesian (**left**) and normalized polar (**right**) coordinates.

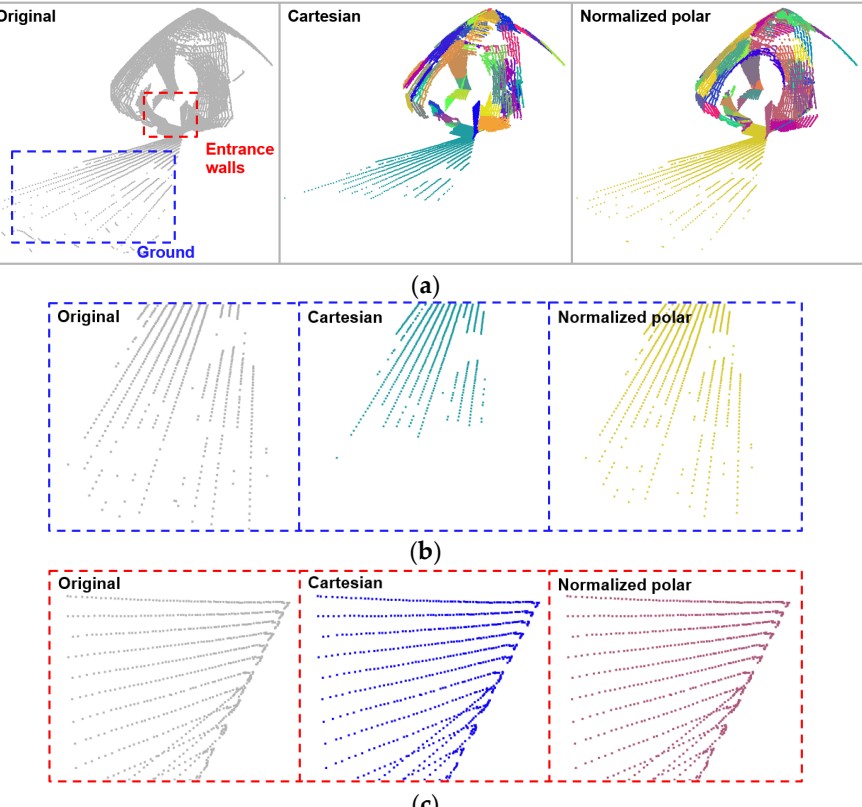

**Figure 24.** Planar feature segmentation results: (**a**) comparison of segmentation results based on Cartesian and normalized polar coordinates of the original point cloud (ground and entrance walls are highlighted in blue and red dotted boxes, respectively), (**b**) zoom-in window of the ground feature, and (**c**) zoom-in window of the entrance wall feature.

### 3.3.3. LSA Using Roof, Stringer, and Wall/Ground Features

In this last step, we perform the full fine registration procedure using all available features (i.e., roof, stringers, and planar walls/ground). This section is dedicated for showing the constraint equations used in the LSA optimization procedure. As the constraint equations for the dome surface have been already introduced (Equation (2)), the stringer and planar feature constraints will be introduced below.

The parametric model for the stringer includes the position of its apex $(X_0, Y_0, Z_0)$, coefficients of the quadratic curve $(a, b)$, and an angular parameter $(\alpha_s)$ that describes the location of the stringer along the dome surface. One should note that the first two sets of parameters are the same as those for the roof surface. The angular stringer parameter is schematically illustrated in Figure 25. As shown in this figure, a stringer-specific coordinate system $(X_s, Y_s, Z_s)$ is defined through a clockwise rotation of the dome coordinate system $(X_d, Y_d, Z_d)$, with an $\alpha_s$ rotation angle around the $Z_d$ axis until the stringer in question resides within the rotated $X_d Z_d$ plane (i.e., the $X_s Z_s$ plane). The coordinates of the edge point relative to the stringer-specific coordinate systems is given by Equation (4). Edge points along a stringer must satisfy the constraint in Equation (5). Since all the stringer's edge points are coplanar (i.e., they should be in the $X_s Z_s$ plane), each edge point must also satisfy the constraint in Equation (6). Considering Equations (4)–(6), one can observe that the stringer feature constraints incorporate feature parameters, as well as registration parameters $R_{p(k)}^m$ and $r_{p(k)}^m$, which are embedded in $r_I^m$. It is important to note that with the introduction of stringers, the $k$ rotation angle of the *source* scans can be solved for.

$$r_I^s = R_d^s r_I^d = R_d^s \left( r_I^m - r_{apex}^m \right) \tag{4}$$

where:

- $r_I^s$ is the coordinates of an edge point in the stringer-specific coordinate system $(x_s, y_s, z_s)$;
- $R_d^s$ is the rotation matrix around the $Z_d$ axis, defined by $\alpha_s$ to transform $(x_d, y_d, z_d)$ into $(x_s, y_s, z_s)$;
- $r_{apex}^m$ is the coordinates of the roof apex in the mapping frame $(x_0, y_0, z_0)$;
- $r_I^m$ is the coordinates of an edge point in the mapping coordinate system $(x_m, y_m, z_m)$; refer to Equation (1) in Section 3.2.2.

$$F_{Stringer_1} = z_s - ax_s^2 - bx_s = 0 \tag{5}$$

$$F_{Stringer_2} = y_s = 0 \tag{6}$$

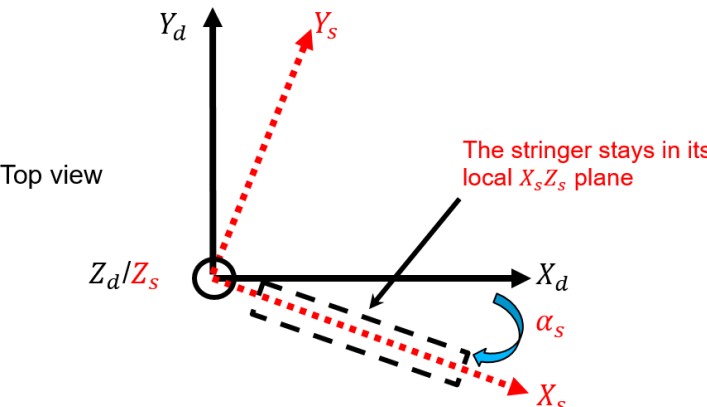

**Figure 25.** Illustration of the stringer-specific coordinate system.

For planar feature constraints, we start by introducing the representation scheme for these features. Based on the main orientation of the normal vector, a planar feature has three normal representation alternatives, i.e., $[w_x, w_y, 1]$, $[w_x, 1, w_z]$, and $[1, w_y, w_z]$, for planes with main normal orientation in the *Z*, *Y*, and *X* axis, respectively, as shown in Figure 26. A planar feature has three unknowns: two components for the normal vector—either $(w_x, w_y)$, $(w_x, w_z)$, or $(w_y, w_z)$—and one position component, $(D)$. Each planar point provides one constraint in the LSA, as presented by Equation (7) for a planar feature with its normal vector mainly oriented along the *Z* axis. Considering Equations (1) and (7), it is evident that the planar feature constraint equation incorporates the feature parameters, as well as the registration parameters $R^m_{p(k)}$ and $r^m_{p(k)}$.

$$F_{plane} = w_x x_m + w_y y_m + D - z_m = 0 \qquad (7)$$

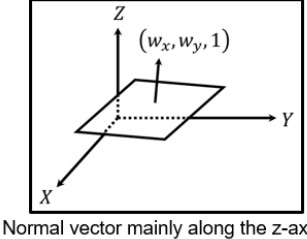 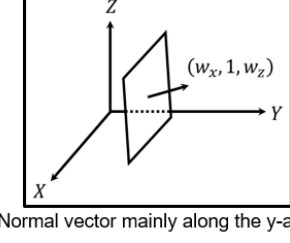 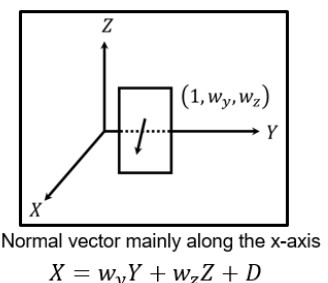

Normal vector mainly along the z-axis    Normal vector mainly along the y-axis    Normal vector mainly along the x-axis

$$Z = w_x X + w_y Y + D \qquad Y = w_x X + w_z Z + D \qquad X = w_y Y + w_z Z + D$$

**Figure 26.** Three forms for representing a plane based on the orientation of the normal vector.

## 4. Experimental Results

This section evaluates the capability of the proposed feature-based registration framework for generating well-aligned point clouds from the SMART system at dome facilities. The conducted experiments address two objectives, namely, (i) evaluate the ability of the roof feature in realizing partial fine registration and (ii) evaluate the comparative performance of different feature combinations (e.g., roof, roof stringers, ground, and/or walls) in achieving full fine registration.

Before presenting the experimental results, the availability of different features for each dataset is presented in Table 2, which shows that the roof feature can be identified in all datasets, regardless of the SMART system mounting configuration (i.e., tripod or roof mounted) or its location within the facility (i.e., close to the dome's center or entrance). In contrast, planar features can be reliably identified only when the SMART system is located close to the facility's entrance (e.g., the *Frankfort* dataset). While the roof feature is visible in the *Frankfort* dataset, the stringers are not clearly visible due to the uneven point density distribution. For the *Lebanon* and *West Lafayette* datasets, the stringers are visible.

**Table 2.** Availability of registration features(shown in red) in the *Lebanon*, *Frankfort*, and *West Lafayette* datasets.

| Salt Dome Facility | | *Lebanon* | | *Frankfort* | | *West Lafayette* | |
|---|---|---|---|---|---|---|---|
| **SMART System Information** | | **Mount** | **Location** | **Mount** | **Location** | **Mount** | **Location** |
| | | **Roof** | **Center** | **Tripod** | **Entrance** | **Tripod** | **Center** |
| **Feature availability** | **Roof** | Visible | | Visible | | Visible | |
| | **Stringers** | Visible | | Not clearly visible | | Visible | |
| | **Ground** | Not visible | | Visible | | Visible in most scans | |
| | **Walls** | Not visible | | Visible | | Visible in most scans | |

### 4.1. Partial Fine Registration

To evaluate the capability of the roof feature in realizing the partial fine registration, we present the LSA results for the three datasets. The partial fine registration results are qualitatively evaluated using vertical slices through the dome roof and heat maps showing the normal distances between the roof feature points and the fitted quadratic surface. Then, quantitative evaluation is conducted by analyzing the quality of fit for the roof and ground features, if available, before/after the partial fine registration (one should note that the partial fine registration is only conducted using the roof feature).

The extracted roof feature from the combined scans of the three datasets is shown in Figure 27, with the point clouds colored by the scan ID. While a uniform point distribution can be observed over the entire roof for the *Lebanon* and *West Lafayette* datasets, a portion of the roof is not visible in the *Frankfort* dataset. The partial visibility of the roof for the latter is expected given the SMART system location near the entrance and the occlusions caused by stored salt. Such partial visibility and sparse point clouds at the dome apex make the stringers not clearly visible in the captured scans within the *Frankfort* dataset.

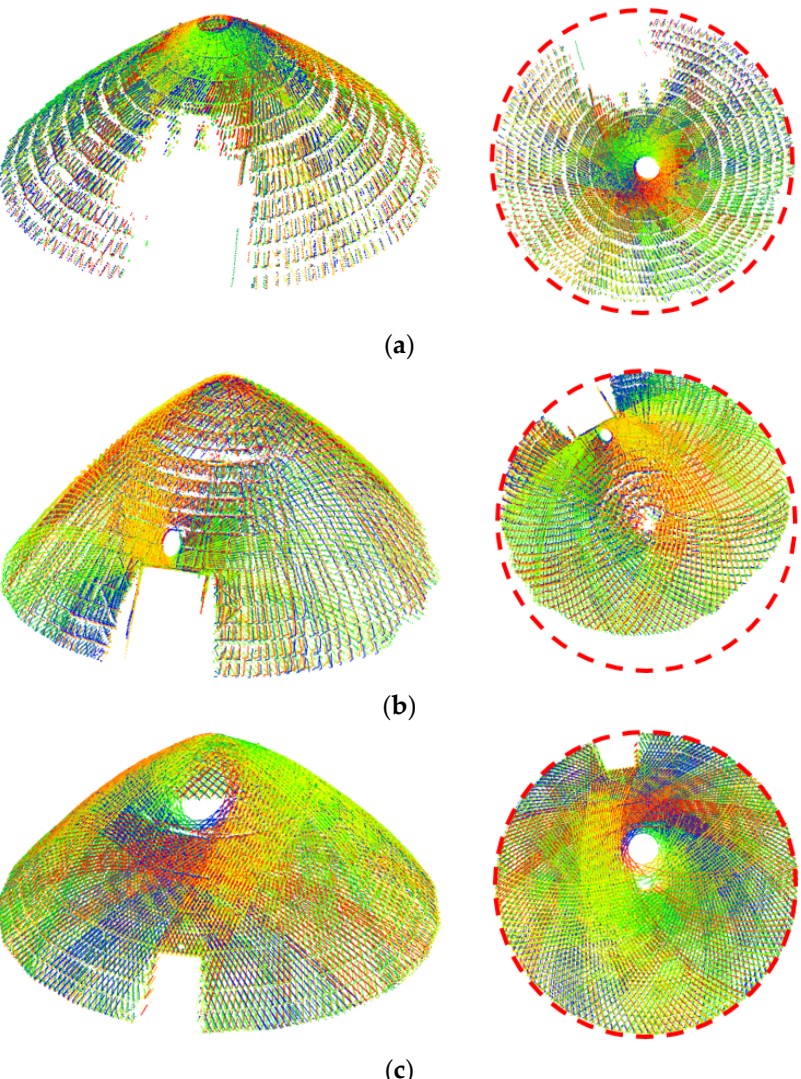

(a)

(b)

(c)

**Figure 27.** Perspective view (**left**) and top view (**right**) of the extracted roof feature for: (**a**) *Lebanon*, (**b**) *Frankfort*, and (**c**) *West Lafayette* datasets (points are colored by scan ID with the red circle showing the outline of the facility).

Extracted vertical profiles from the three datasets before and after the partial fine registration using the roof feature are illustrated in Figures 28–30. As can be seen in these figures, the alignment among scans using the proposed quadratic, surface-based, partial fine registration has been improved compared to that after the image-based coarse registration (e.g., refer to the zoomed-in windows of the profiles for the roof, salt surface, and ground). A closer inspection of these figures reveals that the alignment of the salt surface is slightly worse than the roof and ground. This is attributed to the unsolved $\kappa$ rotation angle, whose impact would only be visible for the salt surface. For further qualitative evaluation, we show heat maps, together with corresponding box plots representing the normal distances between roof feature points and the fitted quadratic surface for these datasets before/after partial fine registration in Figures 31–33. As can be seen in these figures, the quality of fit for the quadratic surface is enhanced after the partial fine registration.

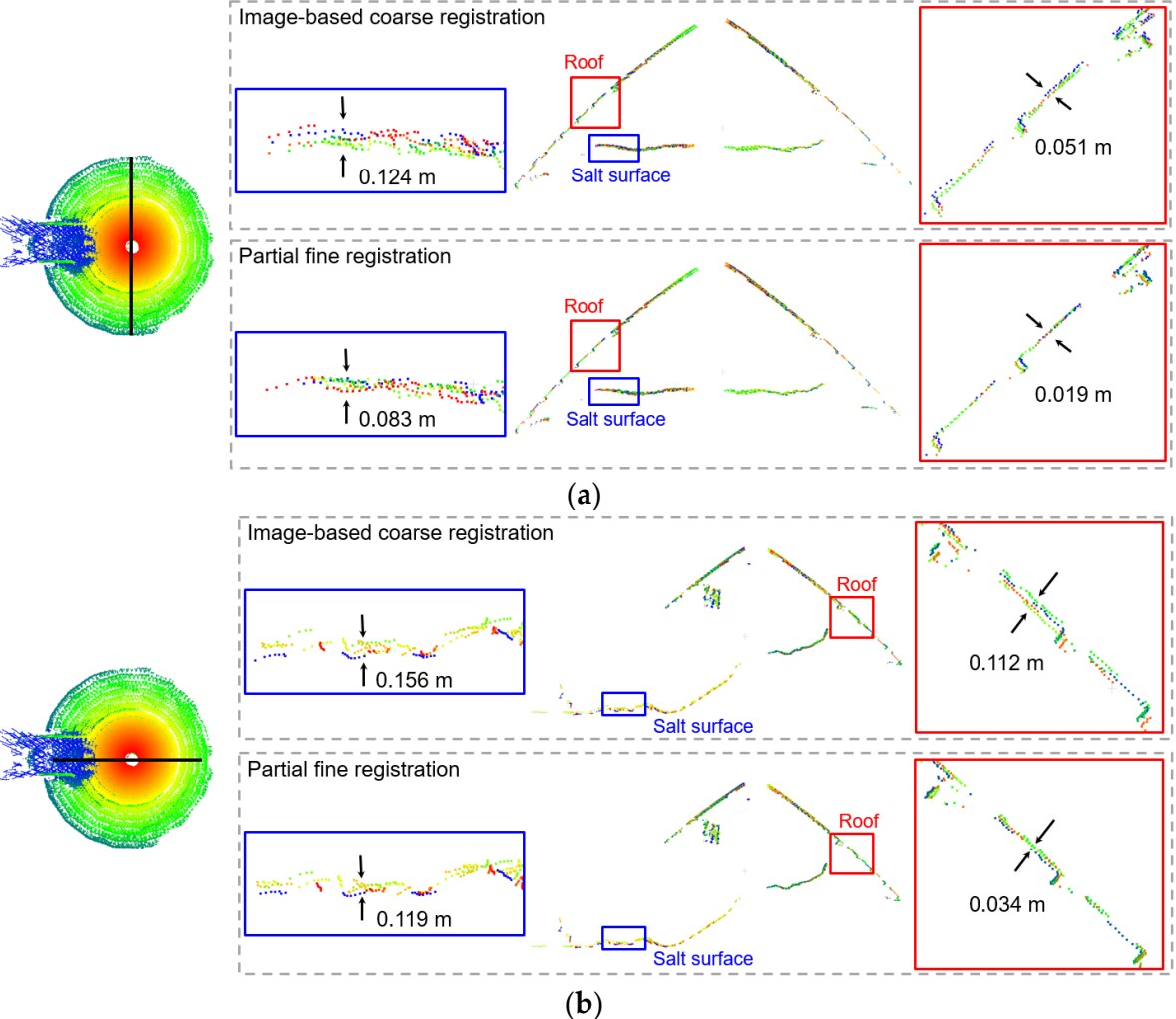

**Figure 28.** Extracted vertical profiles (width: 0.2 m) for the *Lebanon* dataset before/after partial fine registration (the arrows indicate the direction of the measured distance): (**a**) profile orthogonal to the entrance walls and (**b**) profile parallel to the entrance walls.

To quantitatively assess the alignment quality for the point clouds, the Root Mean Square (RMS) of normal distances between the points and their respective best-fitted feature—fitting error—before and after partial fine registration are reported in Table 3. The RMS of the fitting error for the ground feature, if available, is also listed in Table 3 (the ground feature cannot be extracted from the *Lebanon* dataset, as previously reported in Table 2). Compared to the number of points for the ground feature, the large number

of roof points in Table 3 indicates that the roof feature can be reliably identified in the acquired scans. It is evident that the RMS of the fitting error after the image-based course registration is reduced for the three datasets. Given the ±3 cm range noise of the used scanners, the 4-to-6 cm RMS of the fitting error for the roof feature in Table 3 is an indication of the validity of the proposed partial fine registration procedure (i.e., especially when considering the propagation of errors in the pointing direction of the different laser beams, as well as errors introduced by slight deviation of the actual roof feature from a perfect quadratic surface [36]). The slightly higher RMS fitting error for the ground feature (i.e., in the 5-to-7 cm range) is attributed to the facts that it is not used in the partial fine registration and deviation is expected from the planarity of the ground in the dome facility. A close inspection of the roof quality of fit for the *Lebanon* dataset in Table 3 reveals that the roof-mounted SMART system provides better image-based coarse registration results than the other two datasets. Such performance is attributed to the better feature distribution in the captured imagery caused by the smaller SMART-to-object distance for this facility.

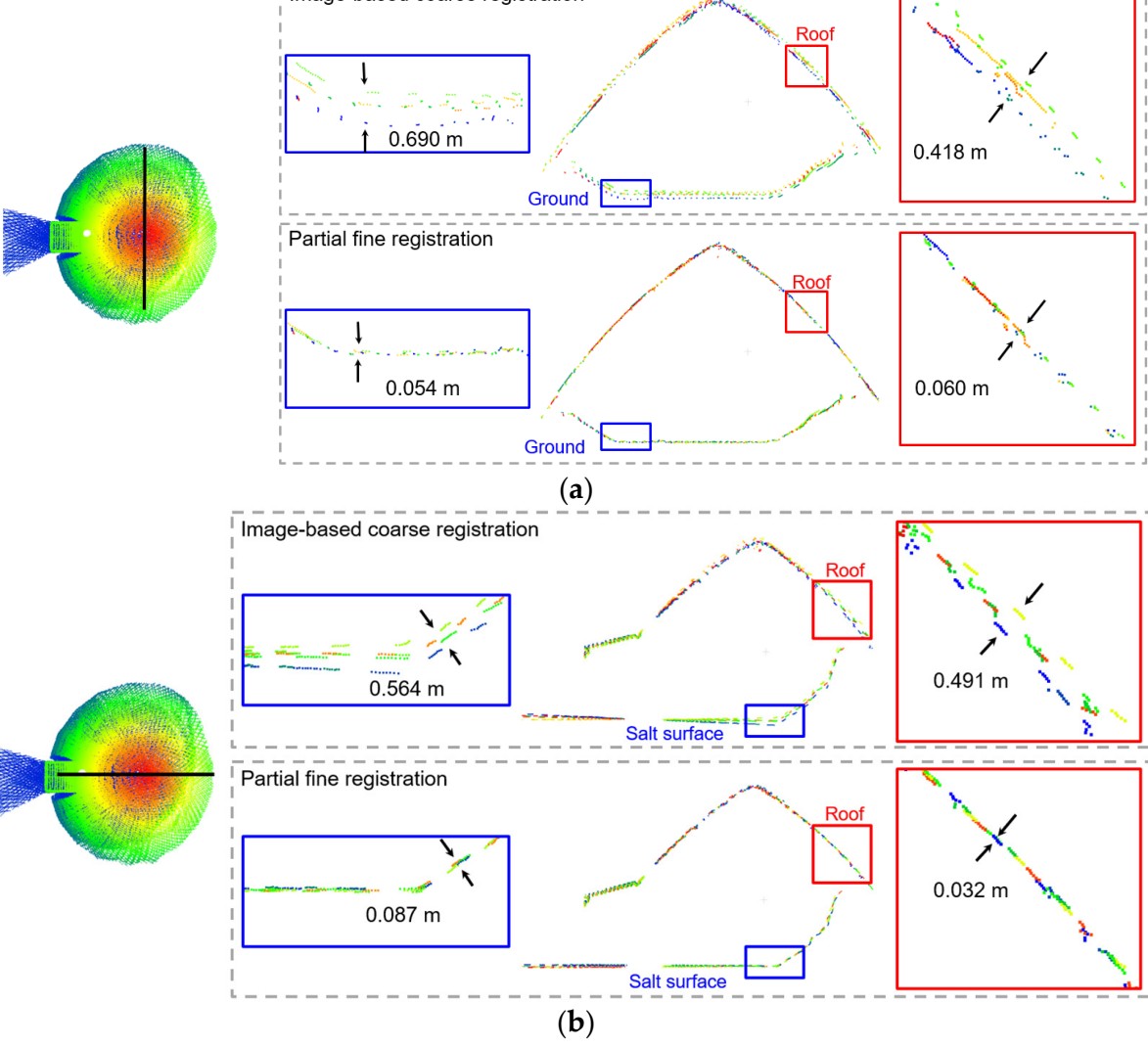

**Figure 29.** Extracted vertical profiles (width: 0.2 m) from the *Frankfort* dataset before/after partial fine registration (the arrows indicate the direction of the measured distance): (**a**) profile orthogonal to the entrance walls and (**b**) profile parallel to the entrance walls.

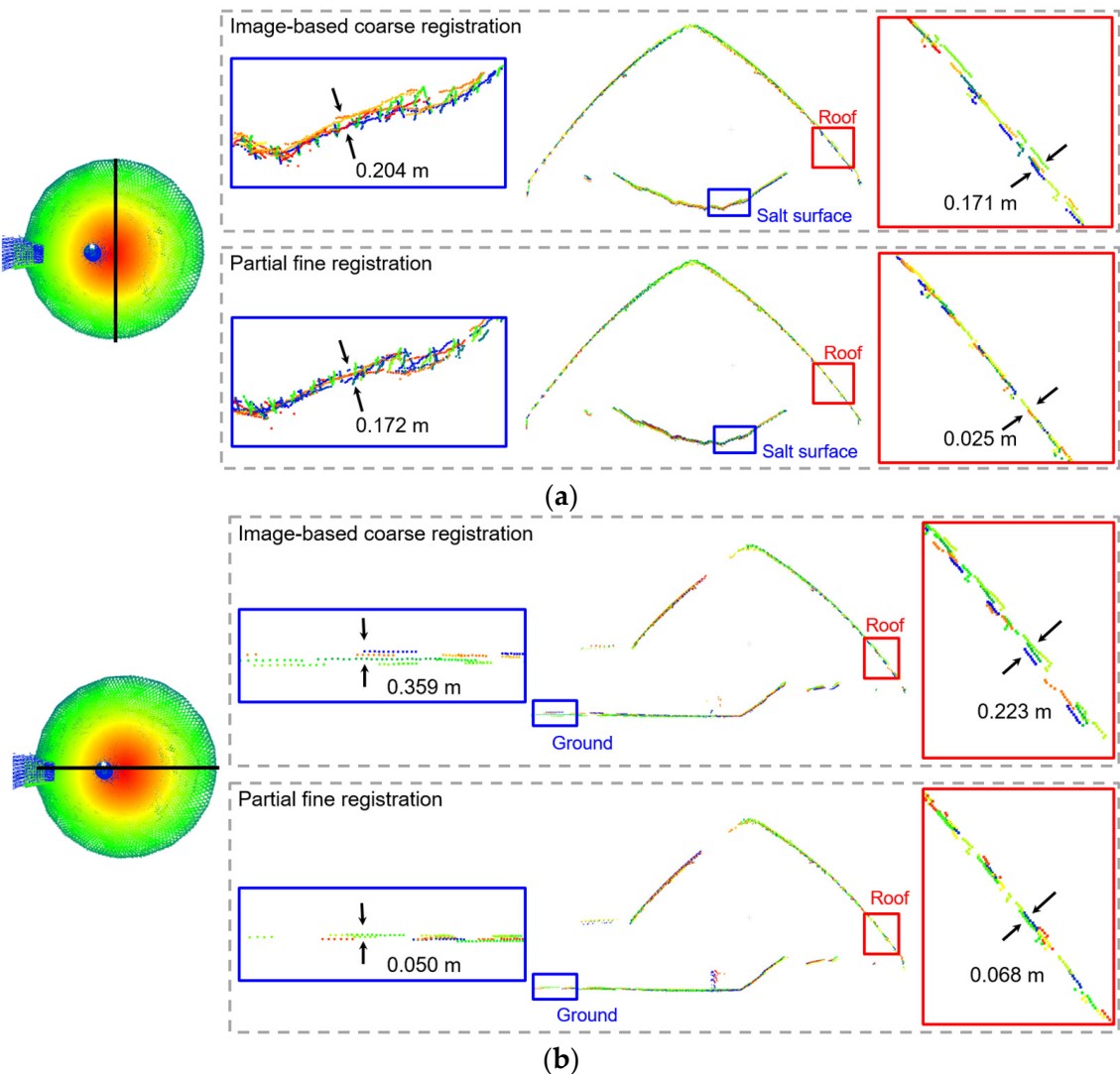

**Figure 30.** Extracted vertical profiles (width: 0.2 m) from the *West Lafayette* dataset before/after partial fine registration (the arrows indicate the direction of the measured distance): (**a**) profile orthogonal to the entrance walls and (**b**) profile parallel to the entrance walls.

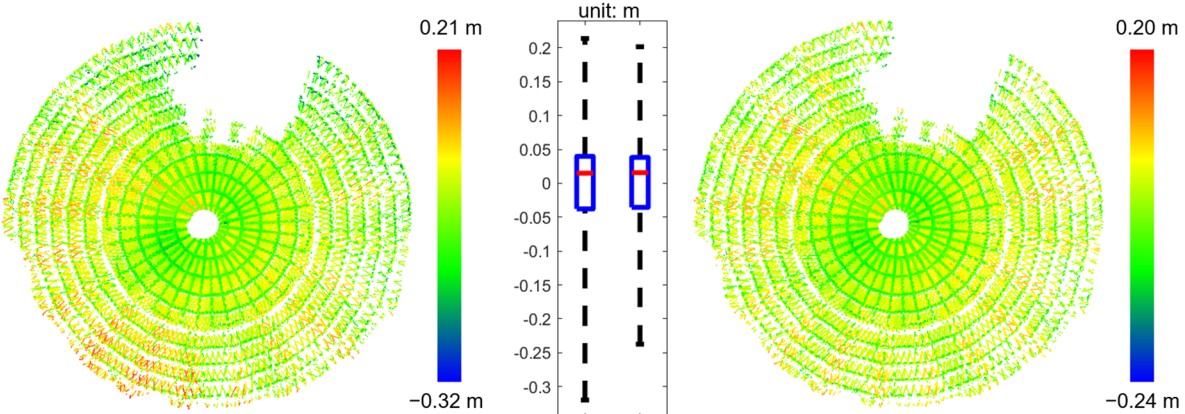

**Figure 31.** Heat maps and box plots representing the normal distance between the roof feature points and fitted quadratic surface for the *Lebanon* dataset before (**left**) and after (**right**) partial fine registration.

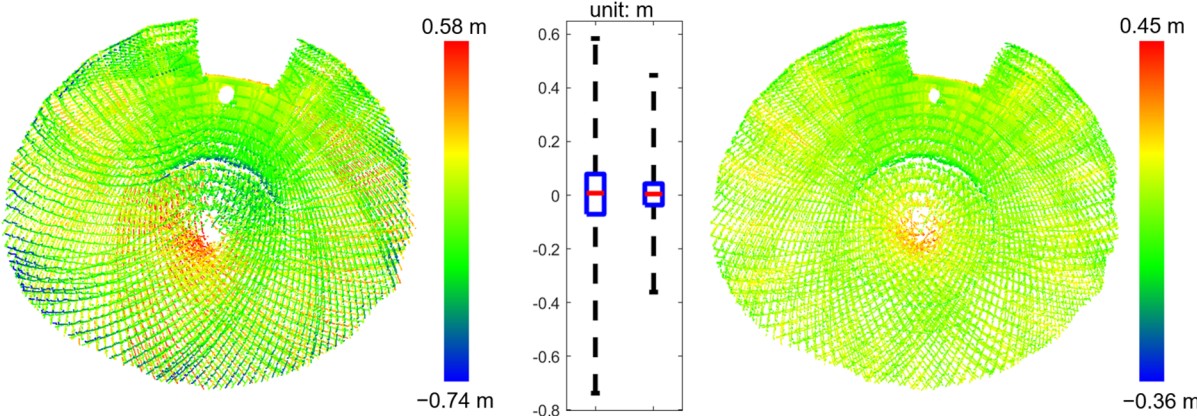

**Figure 32.** Heat maps and box plots representing the normal distance between the roof feature points and fitted quadratic surface for the *Frankfort* dataset before (**left**) and after (**right**) partial fine registration.

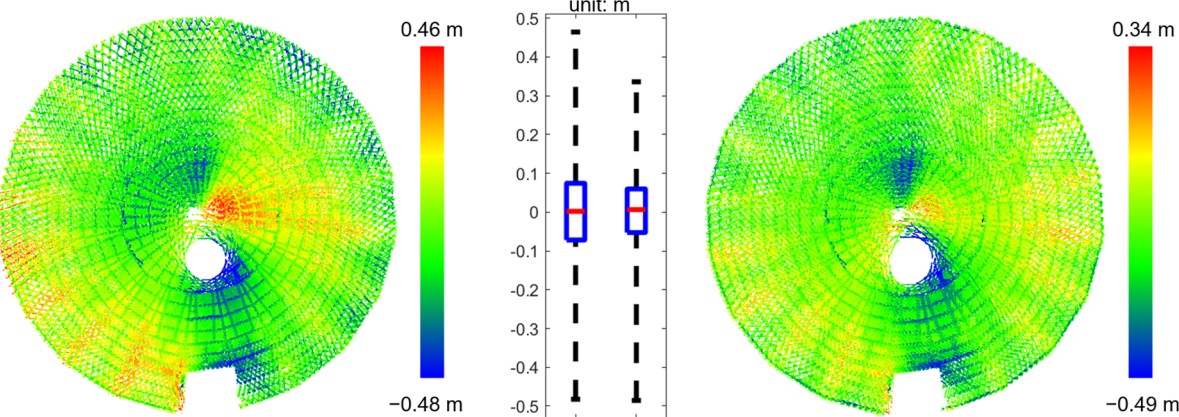

**Figure 33.** Heat maps and box plots representing the normal distance between the roof feature points and fitted quadratic surface for the *West Lafayette* dataset before (**left**) and after (**right**) partial fine registration.

**Table 3.** RMS of the fitting error for the roof and ground features following the coarse/partial fine registration.

| | **Roof Quality of Fit (Utilized in LSA)** | | |
|---|---|---|---|
| **Dataset** | **Number of Points** | **RMS of Normal Distance (m)** | |
| | | **Coarse** | **Partial** |
| *Lebanon* | 339,648 | 0.046 | 0.044 |
| *Frankfort* | 276,769 | 0.095 | 0.051 |
| *West Lafayette* | 415,896 | 0.085 | 0.067 |
| | **Ground Quality of Fit (not Utilized in LSA)** | | |
| **Dataset** | **Number of Points** | **RMS of Normal Distance (m)** | |
| | | **Coarse** | **Partial** |
| *Lebanon* | - | - | - |
| *Frankfort* | 181,846 | 0.229 | 0.071 |
| *West Lafayette* | 115,867 | 0.154 | 0.053 |

### 4.2. Full Fine Registration

This subsection aims at addressing two related objectives, namely, (1) evaluating the capability of using either roof stringers or vertical walls to solve for the $\kappa$ rotation angles of the different scans and (2) assessing the comparative performance of using different feature combinations (e.g., *roof/stringer* vs. *roof/stringer/ground/wall* features). For the first objective, we use the *Lebanon* and *Frankfort* datasets. In the *Lebanon* datasets, roof stringers are used. For the *Frankfort* dataset, on the contrary, vertical walls are used. For the second objective, we conduct a comparative test using the *West Lafayette*, where the full fine registration is achieved using the *roof/stringers* and *roof/stringers/ground/walls* as the registration features. For all the tests, the registration results are qualitatively evaluated using vertical slices through the salt stockpile. Then, a quantitative evaluation is conducted by analyzing the RMS of the fitting error for the utilized features before/after the full fine registration. For the second objective, the salt pile volume for the *West Lafayette* dataset is computed for the different feature primitives.

#### 4.2.1. Lebanon Dataset

As mentioned earlier, the *Lebanon* dataset is collected with a roof-mounted SMART system at the center of the dome. This mounting configuration, together with the relatively full facility, would not allow for scanning the entrance walls and ground. This omission renders the roof and its stringers as the only features for solving the full fine registration. Figure 34 illustrates vertical profiles through the salt surface before and after the full fine registration. As can be seen in this figure, the point cloud alignment at the salt surface has improved after introducing the stringer features for solving the $\kappa$ rotation angles. For an easier-to-interpret visualization, top views of extracted edge points representing the stringers in the scans before and after the full fine registration are shown in Figure 35. The improvement can be seen in the zoom-in windows in this figure (one should note that the stringer width in this facility is approximately 0.1 m).

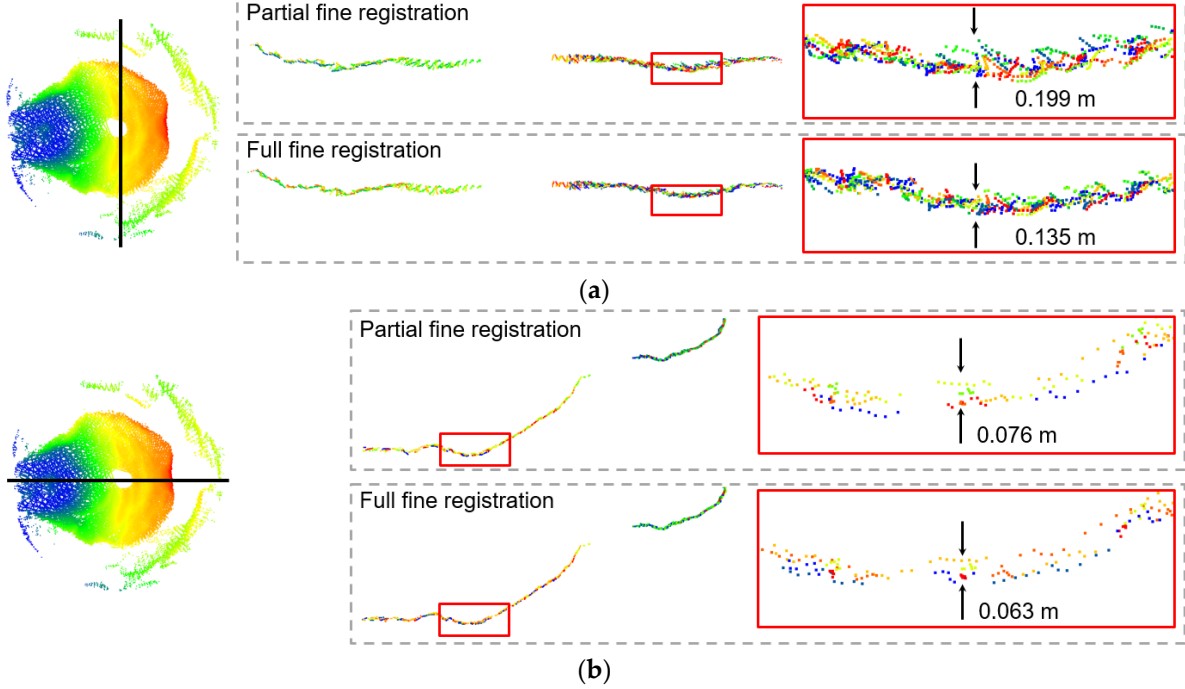

**Figure 34.** Extracted vertical profiles (width: 0.2 m) through the salt surface of the *Lebanon* dataset before/after full fine registration (the arrows indicate the direction of the measured distance): (**a**) profile orthogonal to the entrance walls and (**b**) profile parallel to the entrance walls.

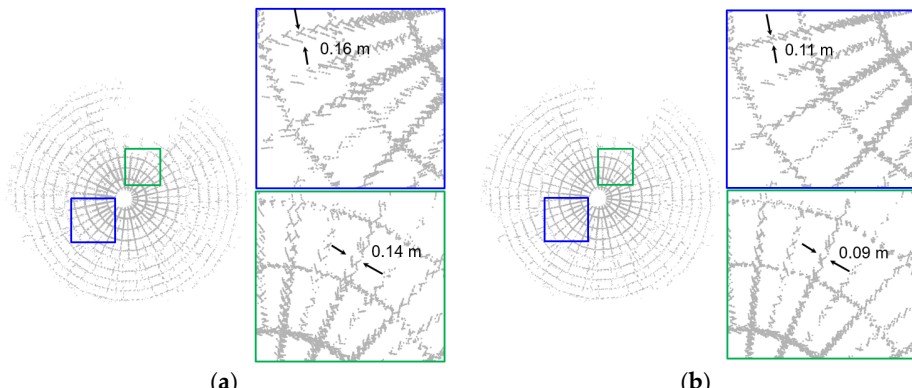

**Figure 35.** Extracted edge points from all scans of the *Lebanon* dataset (**a**) before and (**b**) after full fine registration.

To quantitatively evaluate the impact of adding roof stringers for refining the $\Delta\kappa$ rotation angle, the RMS of the fitting error before and after the full fine registration, along with the number of points in each feature, are presented in Table 4 (the stringer IDs are shown in Figure 36). For the roof feature, we can see that the fitting error remained almost the same before and after the full fine registration. This should be expected because the unresolved $\Delta\kappa$ rotation angle would not impact the fitting error of the roof feature due to its axisymmetric nature. As for the stringers, one can see that their fitting errors reduced from the 9–15 cm range to 7–12 cm. One should note that the fitting error for the stringers is quite acceptable given that the width of the stringer is approximately 10 cm.

**Table 4.** RMS of the fitting error for the registration primitives (*roof* and *stringers*) in the *Lebanon* dataset following the partial/full fine registration.

| Roof Quality of Fit | | |
|---|---|---|
| **Number of Points** | **RMS of Normal Distance (m)** | |
| | **Partial** | **Full** |
| 316,155 | 0.038 | 0.040 |
| **Stringer Quality of Fit** | | |
| **Stringer ID** | **Number of points** | **RMS of Normal Distance (m)** |
| | | **Partial** | **Full** |
| 1 | 1505 | 0.117 | 0.105 |
| 2 | 1440 | 0.121 | 0.109 |
| 3 | 1286 | 0.122 | 0.110 |
| 4 | 1123 | 0.129 | 0.112 |
| 5 | 1096 | 0.130 | 0.114 |
| 6 | 1075 | 0.146 | 0.109 |
| 7 | 1030 | 0.113 | 0.081 |
| 8 | 867 | 0.098 | 0.070 |
| 9 | 682 | 0.093 | 0.069 |
| 10 | 852 | 0.115 | 0.089 |
| 11 | 487 | 0.124 | 0.112 |
| 12 | 1485 | 0.134 | 0.124 |
| 13 | 1464 | 0.121 | 0.113 |
| 14 | 1346 | 0.111 | 0.102 |
| 15 | 1478 | 0.106 | 0.098 |
| 16 | 1536 | 0.102 | 0.093 |
| 17 | 1322 | 0.092 | 0.082 |
| 18 | 1194 | 0.090 | 0.081 |
| 19 | 995 | 0.096 | 0.085 |
| 20 | 1245 | 0.108 | 0.096 |

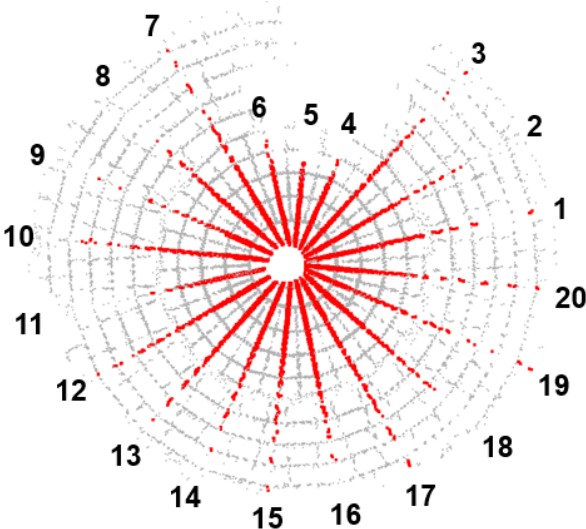

**Figure 36.** Extracted stringer features and the corresponding feature IDs for the *Lebanon* dataset.

### 4.2.2. Frankfort Dataset

The *Frankfort* dataset is collected with a tripod-mounted SMART system stationed near the entrance of the facility. For this dataset, the roof/ground/wall features are visible in all the scans, while the stringer features are not clearly visible. It should be noted that the configuration of the wall features (i.e., parallel entrance walls) do not provide control for evaluating the planimetric shift in the direction parallel to their planes. Therefore, for the *ground/wall*-based registration, we only solve for one of the planimetric shift components among the scans in the registration process (i.e., for the un-resolved planimetric shift component, we rely on that derived from the images-based coarse registration). Therefore, the registration using *wall/ground* features is denoted as another partial fine registration approach (partial 2). Only after adding the roof feature can we conduct the full fine registration. Figure 37 shows extracted vertical profiles along the salt surface before and after the partial (using either *roof* or *ground/wall* features) and full fine registration using *roof/ground/wall* features. A close inspection of the zoomed-in regions in that figure reveals the improvement in the registration after including the *roof* feature. For another qualitative assessment, top views of extracted edge points in all the scans are shown in Figures 38 and 39, with the former highlighting the alignment in the direction normal to the stringers—tangential direction—while the latter shows the alignment in the radial direction. These figures show significant improvement in the tangential and radial directions following the addition of the *roof* features.

Table 5 reports the RMS of the fitting errors after partial fine registration using the *roof* feature, partial fine registration using the *ground/wall* features, and full fine registration using *roof/ground/wall* features. By inspecting the numerical values for the *roof* feature in this table, one can see that the partial fine registration and full fine registration using the *roof* and *roof/ground/wall* features show similar alignment quality (roughly 5 cm). The partial fine registration using the *ground/wall* features, on the contrary, produces more misalignment in the range of 9 cm. This should be expected, as the *ground/wall* features do not provide control for removing misalignment in the direction parallel to the entrance walls. For the *ground* and *wall* features, both the partial fine registration using *ground/wall* features and full fine registration using *roof/ground/wall* features produce comparable alignment quality, which is also expected.

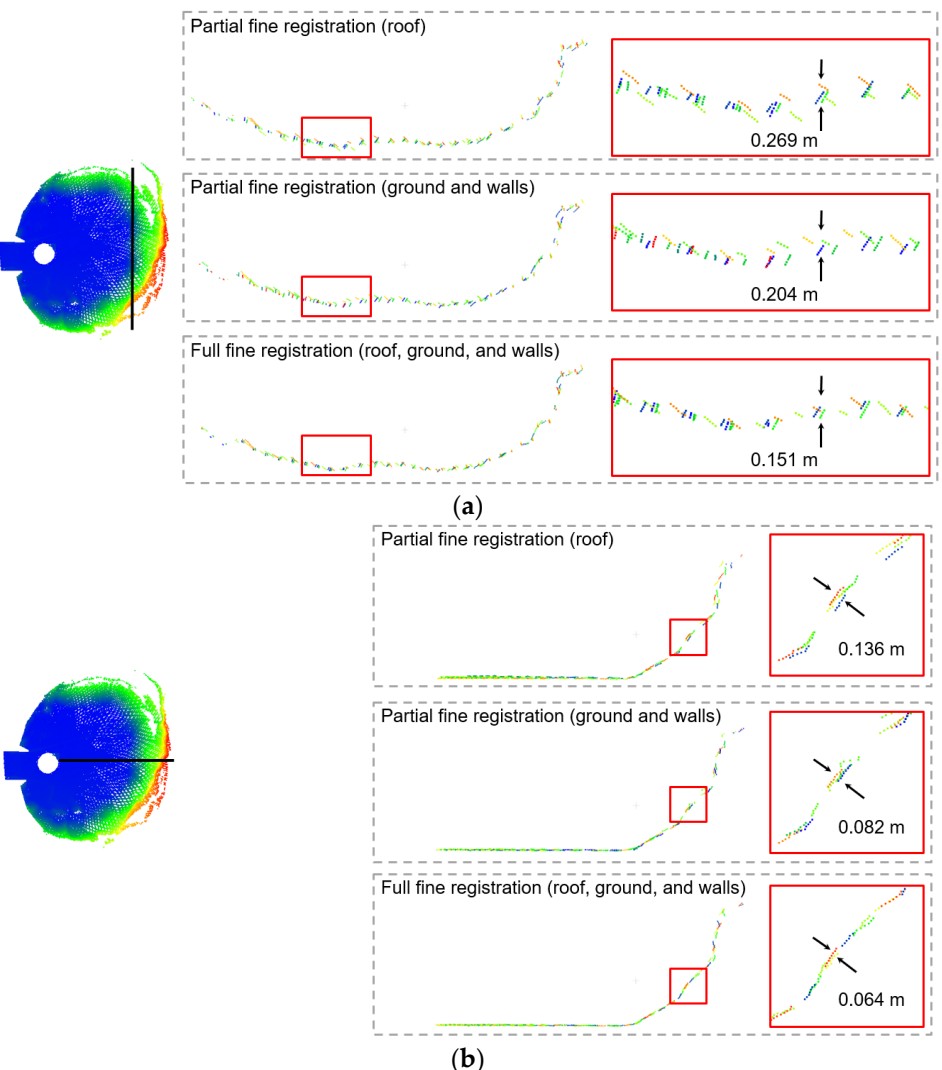

**Figure 37.** Extracted vertical profiles (width: 0.2 m) from the salt surface of the *Frankfort* dataset after the partial fine registration using the *roof* feature, partial fine registration using the *ground/wall* features, and full fine registration using the *roof/ground/wall* features (the arrows indicate the direction of the measured distance): (**a**) profile orthogonal to the entrance walls and (**b**) profile parallel to the entrance walls.

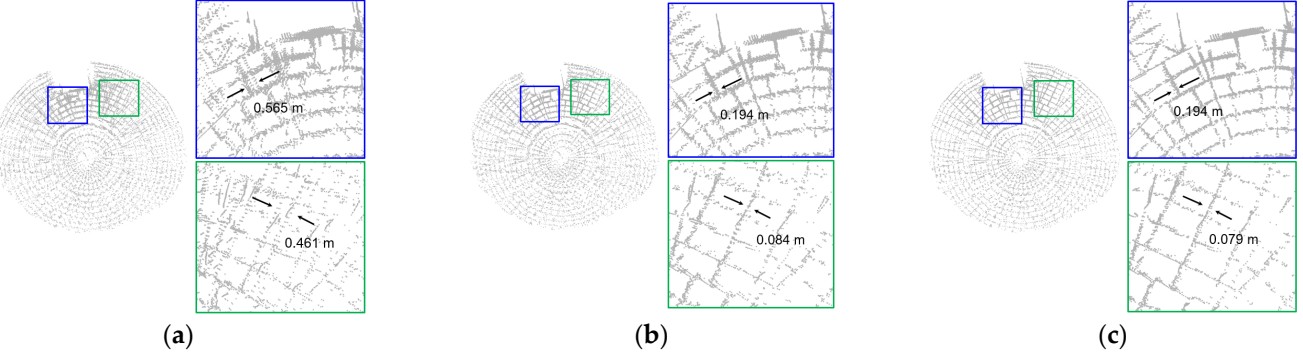

**Figure 38.** Extracted edge points from the scans in the *Frankfort* dataset showing the tangential alignment of the radial stringers after the: (**a**) partial fine registration using the *roof* feature, (**b**) partial fine registration using the *ground/wall* features, and (**c**) full fine registration using the *roof/ground/wall* features.

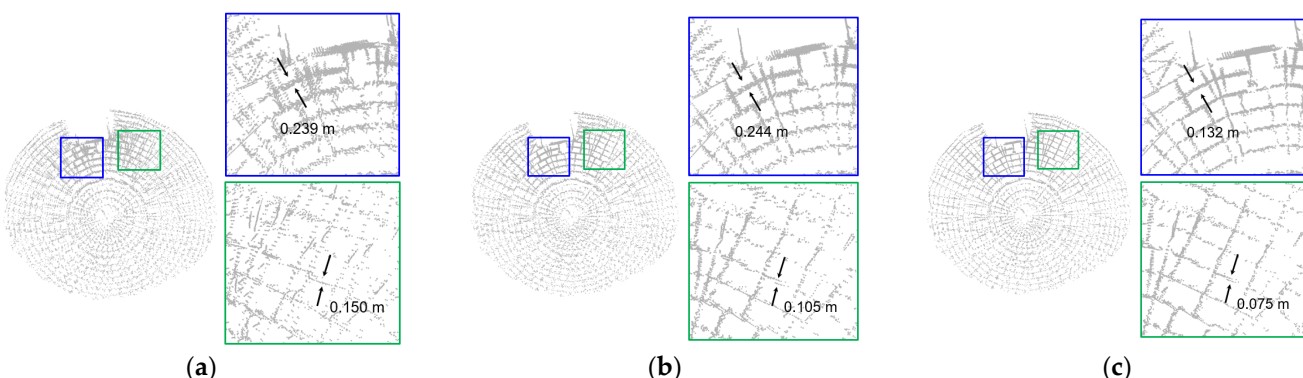

(a)         (b)         (c)

**Figure 39.** Extracted edge points from the scans in the *Frankfort* dataset showing the radial alignment of the stringer rings after the: (**a**) partial fine registration using the *roof* feature, (**b**) partial fine registration using the *ground/wall* features, and (**c**) full fine registration using the *roof/ground/wall* features.

**Table 5.** RMS of the fitting error for the registration primitives in the *Frankfort* dataset after partial fine registration using the *roof* feature (partial 1), partial fine registration using the *ground/wall* (partial 2) features, and full fine registration using the *roof/ground/wall* features.

| | Roof Quality of Fit | | |
|---|---|---|---|
| **Number of Points** | **RMS of Normal Distance (m)** | | |
| | **Partial 1** | **Partial 2** | **Full** |
| 276,769 | 0.051 | 0.086 | 0.052 |
| | **Ground Quality of Fit** | | |
| **Number of Points** | **RMS of Normal Distance (m)** | | |
| | **Partial 1** | **Partial 2** | **Full** |
| 181,846 | 0.071 | 0.033 | 0.033 |

| | **Vertical Walls Quality of Fit** | | | | | | |
|---|---|---|---|---|---|---|---|
| **Left** | | | | **Right** | | | |
| **Number of Points** | **RMS of Normal Distance (m)** | | | **Number of Points** | **RMS of Normal Distance (m)** | | |
| | **Partial 1** | **Partial 2** | **Full** | | **Partial 1** | **Partial 2** | **Full** |
| 34,951 | 0.190 | 0.046 | 0.047 | 26,688 | 0.220 | 0.047 | 0.048 |

### 4.2.3. West Lafayette Dataset

The *West Lafayette* dataset is collected with the tripod-mounted SMART system located somewhere between the center of the facility and its entrance. As a result, the vertical walls at the entrance are visible in the majority of the captured scans. The relatively small amount of salt stored in this facility allowed for the acquisition of point clouds covering the ground. Considering the availability of various features (i.e., *roof/stringers/ground/walls*), a comparative test is conducted to investigate whether the *roof/stringer* features are sufficient for conducting the full fine registration. More specifically, this dataset is processed with the following feature combinations: *roof/stringers* and *roof/stringers/ground/walls*. Figure 40 illustrates extracted vertical profiles along the salt surface before and after the full fine registration. Through this figure, one can see an improvement in the salt surface alignment using either feature combinations, with the *roof/stringers/ground/walls* features showing slightly better quality compared to that using the *roof/stringer* features. As a further qualitative evaluation, top views of extracted edge points from all scans before and after the full fine registration are shown in Figure 41, which shows a similar pattern of improvement when using either the *roof/stringer* or *roof/stringer/ground/wall* features.

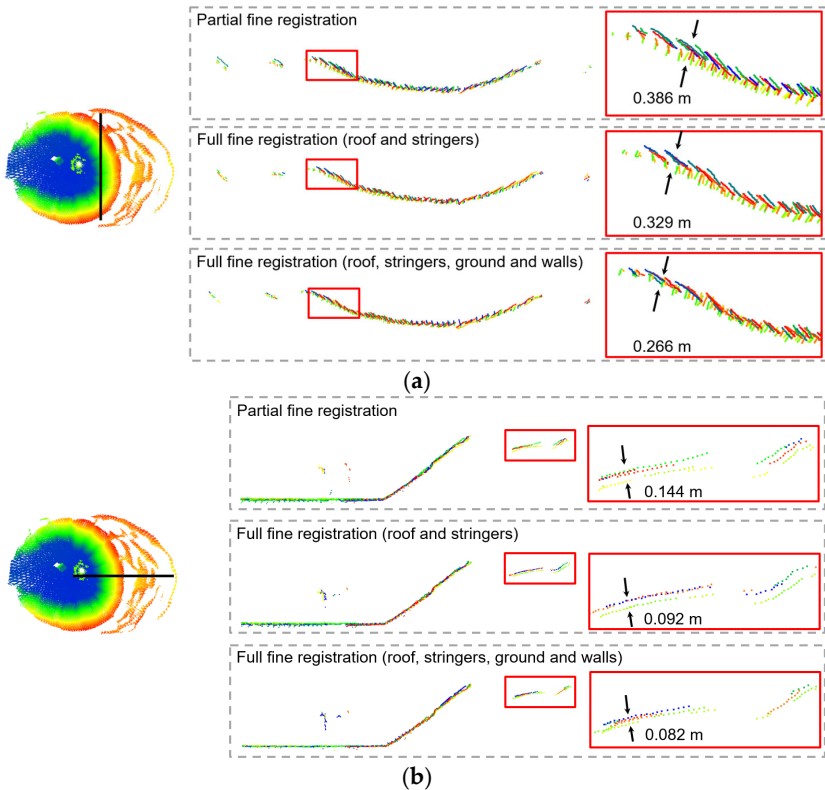

**Figure 40.** Extracted vertical profiles (width: 0.2 m) along the salt surface of the *West Lafayette* dataset before/after the full fine registration (the arrows indicate the direction of the measured distance): (**a**) profile orthogonal to the entrance walls and (**b**) profile parallel to the entrance walls.

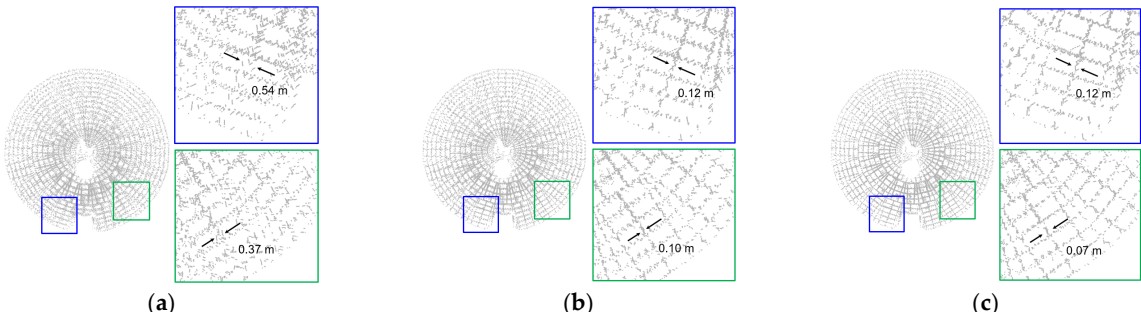

**Figure 41.** Extracted edge points from the scans in the *West Lafayette* dataset before/after the full fine registration: (**a**) partial fine registered edge points, (**b**) full fine registered edge points using *roof/stringer* features, and (**c**) full fine registered edge points using the *roof/stringer/ground/wall* features.

The initial/final RMS of the fitting errors for the *roof/stringer* and *roof/stringer/ground/wall* features are presented in Table 6. While a total of 69 stringers were detected/used for the full fine registration process, the fitting errors for only 20 stringers are listed in Table 6 (refer to Figure 42). In general, the fitting quality for the stringers has improved after the full fine registration for both feature combinations, while showing similar improvement magnitude. Some abnormally high fitting errors can be seen in Table 6 (shown in red). The large fitting errors for these stringers are mainly caused by some outliers. As can be seen in Figure 42, the stringer features in the *West Lafayette* dataset are much closer to each other, with an average angular separation of ~3°, than those for the *Lebanon* dataset. In spite of some outliers, the large number of inlier stringers would still ensure the alignment quality of the final point cloud. The reported values for the *ground/wall* features in Table 6 show good feature alignments in both full fine registration results, while the second combination

provide better alignment (this is expected due to the explicit use of such features in the full fine registration). Finally, a volume estimation is conducted using the derived point clouds from the two feature combinations. The calculated volumes for the *roof/stringer* and *roof/stringer/ground/wall* feature combinations are 1194.069 m$^3$ and 1209.101 m$^3$, respectively, with a difference percentage of 1.24% that can be considered insignificant. In summary, qualitative and quantitative evaluations indicate that the roof and stringer features are sufficient for conducting the full fine registration. The ability to use these features is quite beneficial for situations where *ground/wall* features are not visible due to the SMART mounting/location within the facility, as well as due to a relatively large amount of salt.

**Table 6.** RMS of the fitting error of the registration primitives in the *West Lafayette* dataset after partial fine registration using the *roof* feature, full fine registration using the *roof/stringer* features (Full 1), and full fine registration using the *roof/stringer/ground/wall* features (Full 2).

| Roof Quality of Fit | | | |
|---|---|---|---|
| **Number of Points** | **RMS of Normal Distance (m)** | | |
| | **Partial** | **Full 1** | **Full 2** |
| 403,158 | 0.066 | 0.067 | 0.072 |

| Stringer Quality of Fit | | | |
|---|---|---|---|
| **StringerID** | **Number of Points** | **RMS of Normal Distance (m)** | |
| | | **Partial** | **Full 1** | **Full 2** |
| 1 | 314 | 0.145 | 0.094 | 0.100 |
| 2 | 350 | 0.148 | 0.103 | 0.117 |
| 3 | 302 | 0.165 | 0.133 | 0.143 |
| 4 | 288 | 0.188 | 0.143 | 0.156 |
| 5 | 323 | 0.209 | 0.188 | 0.191 |
| 6 | 212 | 0.208 | 0.195 | 0.210 |
| 7 | 80 | 0.124 | 0.115 | 0.118 |
| 8 | 189 | 0.094 | 0.089 | 0.069 |
| 9 | 350 | 0.122 | 0.075 | 0.055 |
| 10 | 248 | 0.108 | 0.091 | 0.062 |
| 11 | 204 | 0.108 | 0.079 | 0.062 |
| 12 | 254 | 0.149 | 0.082 | 0.079 |
| 13 | 324 | 0.139 | 0.096 | 0.094 |
| 14 | 303 | 0.202 | 0.156 | 0.135 |
| 15 | 611 | 0.202 | 0.182 | 0.195 |
| 16 | 615 | 0.190 | 0.182 | 0.185 |
| 17 | 526 | 0.144 | 0.105 | 0.116 |
| 18 | 471 | 0.078 | 0.057 | 0.070 |
| 19 | 644 | 0.084 | 0.053 | 0.065 |
| 20 | 524 | 0.128 | 0.073 | 0.077 |

| Ground Quality of Fit | | | |
|---|---|---|---|
| **Number of Points** | **RMS of Normal Distance (m)** | | |
| | **Partial** | **Full 1** | **Full 2** |
| 115,867 | 0.053 | 0.050 | 0.029 |

| Vertical Walls Quality of Fit | | | | | | | |
|---|---|---|---|---|---|---|---|
| **Left** | | | | **Right** | | | |
| **Number of Points** | **RMS of Normal Distance (m)** | | | **Number of Points** | **RMS of Normal Distance (m)** | | |
| | **Partial** | **Full 1** | **Full 2** | | **Partial** | **Full 1** | **Full 2** |
| - | - | - | - | 915 | 0.126 | 0.031 | 0.013 |

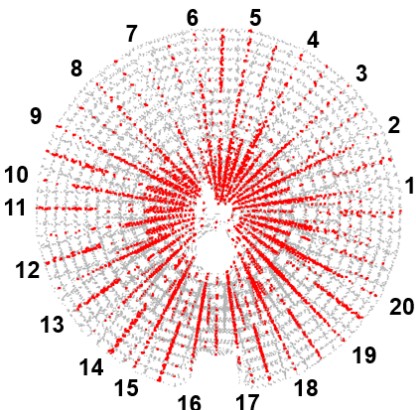

**Figure 42.** Extracted stringer features and the corresponding feature IDs for the *West Lafayette* dataset.

## 5. Conclusions and Recommendations for Future Work

In this study, a feature-based point cloud registration framework has been proposed for deriving well-aligned LiDAR scans acquired inside dome facilities using *Stockpile Monitoring and Reporting Technology* (SMART). Such a development is motivated by the need to produce well-aligned point clouds under challenging conditions, i.e., sparse LiDAR point clouds, signification variation in point density, insufficient overlap among neighboring scans, and unavailability of sufficient planar/linear features. The proposed framework comprises three main steps: image-based coarse registration, partial fine registration using the *roof* feature, and full fine registration using some or all of *roof/stringer/ground/wall* feature combinations. The main contributions of this study can be summarized as follows:

- Two new geometric primitives (*roof* and *stringers* in dome facilities) are investigated as potential features for the registration of collected scans inside dome facilities;
- A semi-automated approach has been developed for the extraction of *roof* and *stringer* features;
- A reliable neighborhood definition approach is developed for extracting planar features from point clouds exhibiting significant variation in point density;
- A general registration framework for processing collected LiDAR data by the SMART system within dome facilities is proposed, while providing the flexibility of including different feature primitives (e.g., roof, stringers, ground, and walls);
- The feasibility of the proposed framework is illustrated using real datasets acquired in three dome facilities.

The results show that the *roof* feature, together with its *stringers*, is sufficient for full fine registration of acquired LiDAR scans inside dome facilities. Using the *roof* and *stringer* features can mitigate the absence of other features (e.g., *ground* and *walls*) due to the mounting/location of the SMART system and/or amount of stored salt. In general, the full fine registration (using more registration primitives) always produces better overall point cloud alignment than partial fine registration (using fewer registration primitives). However, it is worth mentioning that when introducing more registration primitives, the fitting quality for a specific feature might decrease due to the weight reduction of this feature in the registration procedure.

The proposed framework does not incorporate a point-based registration technique due to the low overlap percentage among acquired LiDAR scans. For point clouds collected in storage facilities, a large portion of the point cloud covers the salt surface. This portion is not utilized in the developed procedure. Therefore, future work will augment the developed framework with a point-based registration approach to take advantage of collected points along the stockpile surface. More specifically, overlapping areas along the salt surface among the scans will be investigated to improve the registration quality. In terms of the hardware design, a permanent rail could be installed on the roof for domes with large size

to provide better acquisition of registration primitives, together with sufficient coverage of the salt stockpiles.

**Author Contributions:** Conceptualization, A.H.; formal analysis, investigation, methodology, and validation, J.L., S.M.H., T.Z. and A.H.; software, J.L. and S.M.H.; data curation, R.M.; writing—original draft preparation, J.L. and S.M.H.; writing—review and editing, J.L., S.M.H., T.Z., R.M. and A.H.; supervision, A.H. All authors have read and agreed to the published version of the manuscript.

**Funding:** This work was supported in part by the Joint Transportation Research Program administered by the Indiana Department of Transportation (grant number SPR-4549) and Purdue University. The contents of this paper reflect the views of the authors, who are responsible for the facts and the accuracy of the data presented herein, and do not necessarily reflect the official views or policies of the sponsoring organizations. These contents do not constitute a standard, specification, or regulation.

**Institutional Review Board Statement:** Not applicable.

**Informed Consent Statement:** Not applicable.

**Data Availability Statement:** Data sharing is not applicable to this paper.

**Conflicts of Interest:** The authors declare no conflict of interest.

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
