# Peer review of "An Image-Aided Sparse Point Cloud Registration Strategy for Managing Stockpiles in Dome Storage Facilities"

_remotesensing, doi:10.3390/rs15020504_

Round 1
Reviewer 1 Report
The manuscript presents a study on the volume estimation of stockpiles inside dome storage facilities. The paper describes the application in much detail, with although a bit lengthy could be useful to the reader to follow the logic path-line of your work.
I have the following comments to your work:
Reading at the paper I noticed the similarity of elements of your research with elements in current application in related to survey and shape computation of masonry vaults. For instance, reconstruction of point clouds in narrow areas, elements of shape identification from point clouds, elaboration of data etc. Have a look at the following works in this direction:
- Integrated use of measurements for the structural diagnosis in historical vaulted buildings. Sensors, 20(15), p.4290. https://doi.org/10.3390/s20154290
- A Correspondence Framework for ALS Strip Adjustments based on Variants of the ICP Algorithm. Photogrammetrie - Fernerkundung - Geoinformation Jahrgang 2015 Heft 4 (2015), p. 275 - 289, 2015, DOI: 10.1127/pfg/2015/0270
On the technical side, I was curious to know if you have though also of a permanent system of monitoring for these stockpiles. Clearly using roof and stringer features to mitigate the absence of other features was of help in the present study. However, maybe an automated system based for example on a permanent rail for a LiDAR monitoring system to navigate on the circumferential direction could be a further step to improve your system. Maybe the authors can comment something on this in the paper; probably also in the sense of future work.
Reviewer 2 Report
This study presents an image-assisted fine registration strategy for sparse point clouds acquired in a dome facility, where roof and roof chords are extracted, matched, and modelled as quadratic surfaces and curves. These features are then used in a least squares adjustment (LSA) procedure to obtain well-aligned LiDAR point clouds. Planar features, if available, can also be used in the registration process. The registered point cloud can be used for accurate stack volume estimation. The proposed method was evaluated using datasets obtained from the recently developed camera-assisted LiDAR mapping platform, Stack Monitoring and Reporting Technology (SMART). Experimental results from three datasets show that the proposed method has the ability to acquire well-aligned point clouds within the dome facility with feature fitting errors between 0.03-0.08 m.
The manuscript presents a methodology that I have not seen in other research work and which I consider to be very novel. Also, it is interesting to use LiDAR to scan a large area in all directions for the internal structure and thus perhaps the overall profile of the structure. I believe that the manuscript fits the main thrust of the journal remote sensing and provides a good validation of the proposed method. I therefore recommend that the manuscript be accepted for publication in its current form.
Minor suggestion: some appropriate changes should be made to some of the graphical settings and language to better present the manuscript.
Reviewer 3 Report
Dear authors
I read the article submitted for review with pleasure.
The reviewed manuscript is legible, the designed and processed solution is described in an understandable way, and the entire layout of the article is well thought out and provides the reader with the necessary information. At the same time, the subject matter, the approach to the problem and the proposed solution are interesting from the point of view of the development of modern measurement methods, especially in unusual conditions.
I have no substantive comments to the presented work.
I wish you success in your further work
Reviewer 4 Report
The paper presented a method of LiDAR point-cloud co-registration for monitoring stockpiles in domes. The method is innovative for saving sensor costs by combinations of low-cost sensors instead of high-cost TLS. Also, those are finely described, so I think the manuscript is almost ready for publication. However, I would like the author to address some minor comments as below.
L747: “Some abnormally high fitting errors can be seen in Table 5 (shown in red)” — I think this is a mistake in Table/Figure label as no red fonts in Table 5 and the table is cited under 4.2.2.
Section 5: it is better to mention the results in which full fine registration yields worse RMSE than partial fine registration. This will be helpful for readers to pay attention to better quality control in this method, specifically for busy reader who would skip details of the results.
Thank you in advance for your consideration.
Round 2
Reviewer 1 Report
The manuscript can be accepted in the present form.